# Jump-teaching: Ultra Robust and Efficient Learning with Noisy Labels

## Abstract

Sample selection is the most straightforward technique to combat noisy labels, aiming to prevent mislabeled samples from degrading the robustness of neural networks. However, compounding selection bias and redundant selection operations have always remained challenging in robustness and efficiency. To mitigate selection bias, existing methods utilize disagreement in partner networks or additional forward propagation in a single network. For selection operations, they involve dataset-wise modeling or batch-wise ranking. Any of the above methods yields suboptimal performance. In this work, we propose *Jump-teaching*, a novel framework for optimizing the typical workflow of sample selection. Firstly, Jump-teaching is the *first* work to discover significant disagreements within a single network between different training iterations. Based on this discovery, we propose a jump-manner strategy for model updating to bridge the disagreements. We further illustrate its effectiveness from the perspective of error flow. Secondly, Jump-teaching designs a lightweight plugin to simplify selection operations. It creates a detailed yet simple loss distribution on an auxiliary encoding space, which helps select clean samples more effectively. In the experiments, Jump-teaching not only outperforms state-of-the-art works in terms of robustness, but also reduces peak memory usage by $0.46\times$ and boosts training speed by up to $2.53\times$. Notably, existing methods can also benefit from the integration with our framework.

## 1 Introduction

Learning with Noisy Labels (LNL) is the most promising technique in weakly supervised learning. Generally, noisy labels stem from mistaken annotations of the dataset, such as in crowd-sourcing (Welinder et al., 2010) and online query (Blum et al., 2003). As accurate annotations of large datasets are a time-consuming endeavor, the existence of noisy labels becomes inevitable. Deep neural networks can easily overfit to noisy labels, which is prone to poor generalization performance (Zhang et al., 2021; Han et al., 2020). Furthermore, the efficiency challenge of LNL is often overlooked in comparison to the robustness problem (Bakhshi and Can, 2024), which is vital in real-time (Mahajan et al., 2018; Bakhshi and Can, 2024) or edge security scenarios (Aït-Sahalia et al., 2010). Therefore, this work aims at finding an LNL solution characterized by efficiency and robustness.

Recent LNL methods can be categorized into three types: regularization, label correction, and sample selection. Regularization methods focus on crafting noise-robust loss functions (Ghosh et al., 2017; Wang et al., 2019) and regularization techniques (Liu et al., 2020; Zhang et al., 2020; Cao et al., 2021), but they cannot fully avoid fitting to noisy labels during training, resulting in sub-optimal outcomes. Label correction, integrating closely with semi-supervised learning, aims to refine or recreate pseudo-labels (Han et al., 2019; Sohn et al., 2020; Pham et al., 2021). These methods make use of corrected noisy labels but require computational resources for the estimation of noise transition matrix (Goldberger and Ben-Reuven, 2022) or the ensemble prediction (Lu and He, 2022).

Sample selection is a direct approach for combating label noise (Song et al., 2019; Kim et al., 2021; Wu et al., 2020; Malach and Shalev-Shwartz, 2017; Xia et al., 2023). It typically operates in an iterative workflow, selecting possibly clean samples through a certain process and then updating the parameters based on those samples. However, compounding selection bias and redundant selection operations hinder the effectiveness of workflow.

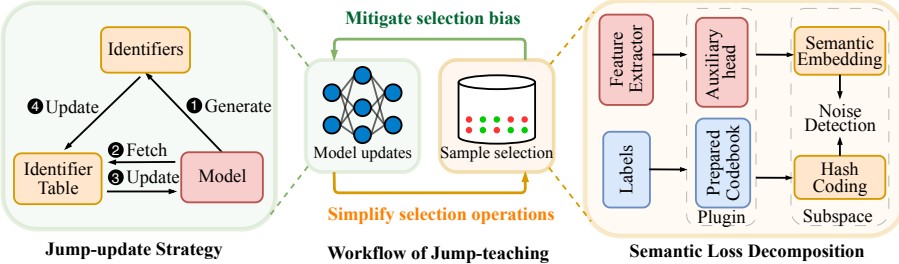

Figure 1: The overview of Jump-teaching. Motivated by two challenges in the sample selection approach, Jump-teaching proposes a novel model-update strategy termed *Jump-update* to mitigate the selection bias and a lightweight sample-selection plugin termed *Semantic Loss Decomposition* to simplify selection operations.

Firstly, selection bias inevitably arises from the exposure of the classifier to noise. This bias causes noisy samples to be included in the training data. When the neural network trains on these data, the error erodes the robustness of the network, thus the bias is amplified. To mitigate this bias, several approaches have been proposed, including Decoupling (Malach and Shalev-Shwartz, 2017), Co-teaching (Han et al., 2018), and Co-teaching+ (Yu et al., 2019), among others. These methods follow a similar paradigm: a partner network is integrated to give different predicted labels from the original network. These disagreements, referring to the existence of differences of networks in the selection behaviors, seek to diverge in each sample selection and guide the training process by model update. However, the extra network commonly used in such methods requires double memory and computation resources. Notably, some methods that use a single network implicitly mitigate the bias by reducing the frequency of conducting selections, but they require additional forward passes through the entire dataset before training begins, which increases computational cost (Yuan et al., 2023; Wu et al., 2020).

Secondly, current selection operations are redundant. The redundancy is evident in the repeated need to aggregate amounts of data for batch processing, which leads to inefficiencies. Specifically, some methods (Han et al., 2018) rely on a small-loss criterion that ranks samples within each batch by their loss magnitude, prioritizing those with smaller losses for training. Other methods (Li et al., 2020) regard the sample selection as a binary classification problem, where the distribution of sample losses is modeled by techniques like Gaussian mixture models (GMM). The core reason for this redundancy lies in the limited information provided by the classification head. The loss is typically represented as a floating-point value, calculated from the discrepancy between predicted probabilities and one-hot encoded labels. It fails to provide enough meaningful information to guide effective sample selection. Consequently, redundant operations are required to compensate for this lack of granularity.

In this paper, we propose an ultra robust and efficient framework *Jump-teaching* that optimizes the workflow of the sample selection approach. Its effectiveness comes from two aspects: firstly, it adopts a jump-manner strategy to mitigate selection bias within a single network. Secondly, it designs a lightweight plugin for a simplified sample-wise selection operation. The overview of Jump-teaching is shown in Fig. 1. Jump-update Strategy is motivated by the discovery of significant disagreement between different training iterations of a neural network, excluding neighboring iterations, from a temporal perspective in Fig. 2. Therefore, this intrinsic disagreement enables the strong self-correction of selection bias. Moreover, we design a lightweight plugin for efficient selection. As shown in Fig. 1, it contains an auxiliary head and a prepared codebook, which transform the outputs and labels into semantic embeddings and hash coding, respectively, within an auxiliary space. When training on clean data, the goal of this plugin is to minimize the gap between semantic features and hash coding as close to $0$ as possible. The ideal distribution of elements of this vector is a mean value distribution with a variance of $0$. Conversely, for noisy data, this gap becomes too large to reach $0$, and its distribution is uncertain. According to the aforementioned characteristics, we can easily and effectively distinguish between clean and noisy samples.

In summary, we make the following contributions: (1) We are the first work to discover significant and persistent disagreement within a single network. Based on this discovery, we propose the jump-update strategy for strong self-correction of bias. The strategy enables a single network to surpass the capacity

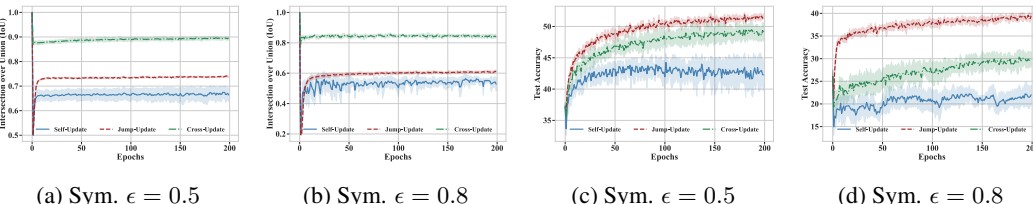

| (a) Sym. $\epsilon = 0.5$ | (b) Sym. $\epsilon = 0.8$ | (c) Sym. $\epsilon = 0.5$ | (d) Sym. $\epsilon = 0.8$ |

Figure 2: "Disagreement" and test accuracy of different strategies with symmetric noise ratio $\epsilon = 0.5$, $\epsilon = 0.8$. We use *Intersection over Union* (IoU) between two selections to represent the disagreement. In the cross-update strategy, the disagreement emerges in a different network, while in the other two strategies, it emerges during different epochs. More details are discussed in Appendix A.1.

of a dual network in the robustness and efficiency of LNL. (2) We propose a lightweight plugin for efficient sample selection. It helps select clean data easily by intrinsic distribution characteristics. (3) Jump-teaching outperforms state-of-the-art methods in robustness across various noise settings, particularly under extreme noise conditions. It also achieves up to $2.53\times$ speedup and reduces peak memory usage by $0.46\times$. (4) The framework is flexible in integrating with other LNL methods, from which both supervised-only and semi-supervised methods benefit.

The rest of this paper is organized as follows. Related work is reviewed in Sec. 2. In Sec. 3, we introduce the proposed framework Jump-teaching. Experiments are illustrated in Sec. 4. Conclusions are given in Sec. 5.

## 2 RELATED WORK

**Sample Selection**. The core idea of the sample selection approach is to filter out noisy samples to prevent the network from fitting to them. Since the loss of a single sample is insufficient for selection, current approaches require additional operations. Most works employ a ranking operation with prior knowledge of the noise ratio, *e.g.*, Co-teaching (Han et al., 2018) and Co-teacing+ (Yu et al., 2019), or probabilistic estimation on loss value, such as Gaussian Mixture Models (GMM) (Permuter et al., 2006) and Beta Mixture Models (BMM) (Ma and Leijon, 2011), while some methods rely on a complex statistical estimation process (Li et al., 2020; Wei et al., 2020; Arazo et al., 2019).

**Sample-selection Bias**. Sample selection inherently involves bias, leading to error accumulation. To avoid this, many methods employ dual networks and correct this bias through disagreement in their selection. Decoupling (Malach and Shalev-Shwartz, 2017) utilizes a teacher model to select clean samples to guide the learning of a student model, Co-teaching (Han et al., 2018) simultaneously trains two networks on data selected by peer network. Co-teaching+ (Yu et al., 2019) maintains divergence between the two networks by limiting the training data to samples where the networks disagree. JoCoR (Wei et al., 2020) trains two networks with the same data and employs regularization to remain divergent. Additionally, many LNL methods integrate the co-training framework to achieve advanced performance (Li et al., 2020; Liu et al., 2020; 2022; Chen et al., 2023). However, these co-training methods significantly double computational overhead. Although some works have utilized predictions in different iterations as selection criteria (Wei et al., 2022; Yuan et al., 2023), no related work has explicitly addressed bias in a single network. Some implicit methods will be elaborated on Sec. 3.

## 3 METHODOLOGY

Jump-teaching is a novel framework for optimizing typical workflow of the sample selection approach. Specifically, it employs *Jump-update Strategy* to mitigate the bias and *Semantic Loss Decomposition* to simplify selection operation. We demonstrate them in Sec. 3.1 and Sec. 3.2, respectively.

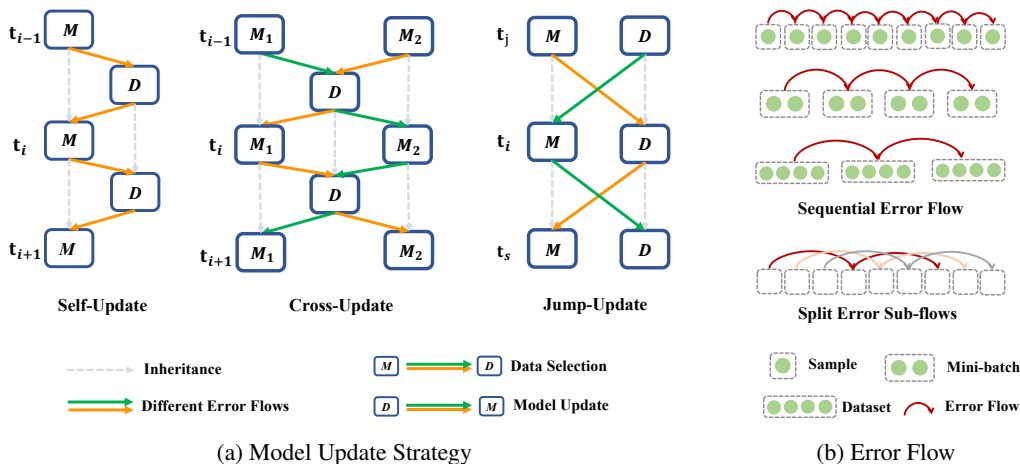

(a) Model Update Strategy        (b) Error Flow

Figure 3: **The Left**: The flow path of error under different update strategies. It leads to varying degrees of accumulation. **The Right**: Two types of error flows. Different from methods that use additional forward passes to reduce accumulation, our strategy splits the error into sub-flows.

### 3.1 JUMP-UPDATE STRATEGY

Inspired by the observation in Fig. 2, we present the Jump-update strategy, a jump-manner model update paradigm. This strategy aims at bridging the disagreement between iterations. First, we provide a detailed description of the strategy. Afterward, we conduct an empirical analysis to explore its debiasing principles and verify our insights in the experimental analysis.

**Strategy Description.** To simplify our discussion, we dive into the procedure of sample selection and give some definitions. Sample selection is accomplished by the execution of model updates and sample selection with multiple iterations. In other words, model updates and sample selection are performed only once in every iteration. Previous strategies and ours are compared in Fig. 3(a). Suppose $t_i$ denotes the current $i$-th iteration, **ancestor iteration** refers to the former iteration $t_j, 0 \le j \le i-2$, excluding the previous $(i-1)$-th iteration, during the procedure of sample selection. Similarly, **descendant iteration** denotes the future iteration $t_s, i+1 \le s \le N_{\text{iterations}}$. $N_{\text{iterations}}$ is the total number of training iterations. The ancestor iteration and descendant iteration will not appear in the same epoch. In our paradigm, the current network in the $i$-th iteration is trained with clean samples selected only by the network from an ancestor iteration $t_j$. This behavior of sample selection exhibits a *jump* form. The name of the algorithm is derived from this point. Concretely, we leverage a binary identifier to represent the outcome of the label judgment after clean sample selection. Thus, a binary identifier table $\mathcal{I}$ corresponds to the entire data. The jump-update strategy is divided into four steps: 1) The neural network in current interaction $t_i$ generates the new binary identifiers by the clean sample selection for the descendant iteration $t_s$. 2) We fetch and cache the old binary identifier table from the ancestor iteration $t_j$. 3) The network updates the parameters based on the clean data judged only by the old table. Before the update, the network inherits the weight from the previous iteration $t_{i-1}$. 4) We utilize the identifiers generated to update the table. When the jump-update strategy is applied, disagreements between different training iterations of a neural network appear and are bridged, leading to a decrease in the accumulated error. Suppose $S$ denotes the jump steps, it should be noted that $2 \le S \le N_{\text{iterations}}$. The evidence of disagreement and the setting of $S$ are illustrated in Sec. 4.2. Moreover, we provide an example for describing this strategy in Appendix A.3.

**Empirical Analysis.** Sample selection inevitably has a bias, leading to accumulated error in an error flow. As shown in Fig. 3(b), the graph of error flow presents a sequential form by iterations while the jump-update strategy splits a sequential error flow into multiple error sub-flows. Intuitively, the more training iterations increase, the more rapidly the error is accumulated. Nonetheless, the degree of error accumulation in the two is significantly different. In the sequential form, errors are accumulated consecutively. As the error sub-flows are orthogonal in the jump-update strategy, each error is accumulated only in its own error sub-flow. Thus, the jump-update strategy has a significantly

smaller degree of accumulated error compared to the sequential form. This is the source of the magic of the jump-update strategy. We formalize the aforementioned procedure mathematically below and give some detailed properties.

Suppose that $N_A$ is the total number of error accumulations, $N_a$ is the number of error accumulations in an error sub-flow, and $N_f$ is the number of error sub-flows. Besides, the constant $e$ represents the number of training epochs and $n$ represents how many selections are made in one epoch. $D_A$ denotes the overall degree of accumulated error, while $d_a^k$ denotes the degree of accumulated error in the $k$-th error sub-flow. In the absence of a specific reference, we can also denote the degree of accumulated error in one error sub-flow by $d_a$.

**Property 1** *The overall degree of accumulated error $D_A$ is proportional to the total number of error accumulation $N_A$, $D_A \propto N_A$. Under the hypothesis that the error flow is an uninterrupted model, the number of error accumulation $N_A$ equals the total number of training iterations $N_{iterations}$, while $N_{iterations}$ equals $e \times n$. Therefore, $D_A \propto n$.*

The overall degree of accumulated error $D_A$ depends on $n$ stated in Property 1. When the network selects data from a mini-batch, $D_A$ can be enormous because $n$ is equal to the number of mini-batches, *e.g.,* Co-teaching. If the network selects data from the entire dataset, $n$ can be reduced to **1**, thereby significantly reducing $D_A$, *e.g.,* TopoFilter (Wu et al., 2020) and LateStopping (Yuan et al., 2023). However, this is inefficient because an additional forward over the entire dataset before training is necessitated. Property 1 is proven in Appendix A.4.

**Property 2** *The accumulated error could be reduced by splitting the error flow into multiple error sub-flows. The degree of accumulated error in the $k$-th error sub-flow $d_a^k$ is proportional to the number of error accumulations in each error sub-flow $N_a$, $d_a^k \propto N_a$. The number of error accumulations in each error flow $N_a$ equals $\frac{N_A}{N_f}$.*

As stated in Property 1, $D_A$ can be reduced by minimizing $n$, while $D_A$ can also be reduced by splitting into error sub-flows as illustrated in Property 2. The first property relates to the sample selection mechanism, and the second property is associated with the different strategies of model updates: 1) The self-update strategy follows a single error flow, resulting in $d_a = D_A$, which leads to rapid error accumulation. 2) The cross-update strategy has two error sub-flows and $d_a = 0.5D_A$, which mitigates the error accumulation to some extent. 3) The jump-update strategy reduces $d_a$ to $0.5e$, which is a significantly minimal value. Further insights are detailed in Appendix A.5.

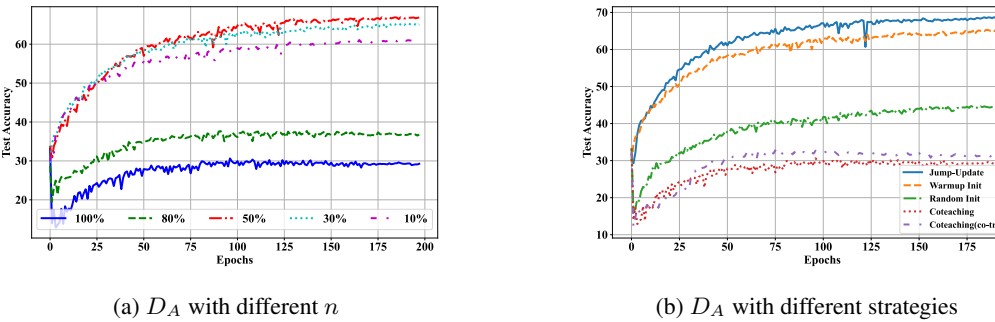

(a) $D_A$ with different $n$           (b) $D_A$ with different strategies

Figure 4: Test accuracies(%) on *CIFAR-10* with Sym. $\epsilon = 0.8$.

**Experimental Analysis.** We verify Property 1 and Property 2 with toy examples, respectively. We employ the small-loss sample selection method from Co-teaching to establish the baselines for self-update and cross-update. Our experiments choose *CIFAR-10* (Krizhevsky et al., 2009) dataset with the symmetric noise ratio $\epsilon = 80\%$. We utilize ResNet-18 (He et al., 2016b) as the backbone and warm up it for one epoch before formal training.

To verify Property 1, we observe $D_A$ by testing the accuracy of the network with different values of $n$. To control $n$, We set $r$ as the proportion of sample selection that takes effects, $D_A$ equals $r \times n$ in this way. We set $r$ to $10\%$, $30\%$, $50\%$, $80\%$, and $100\%$. As shown in Fig. 4(a), rapidly accumulated errors lead to extreme deterioration of the model such as when $r = 100\%$ and $r = 80\%$, while slower

accumulated ones achieve better performance, with a moderate $r = 50\%$ yielding the best results. This is consistent with Property 1.

To verify Property 2, we observe $D_A$ by testing the accuracy of the neural network with three strategies, self-update, cross-update, and jump-update. As shown in Fig. 4(b), cross-update slightly outperforms self-update, while jump-update is significantly more effective than both. Thus, splitting error flows is an effective way to reduce error accumulation. Moreover, we also greatly improved the performance of self-update by reducing $n$. Specifically, we employ two unbiased initial identifier tables, with $r$ set to $30\%$ and $5\%$, respectively. This is discussed in Appendix A.6, which also includes the impact of initial bias.

**Conclusion.** The jump-update strategy is more effective and efficient than previous works. Compared to methods that implicitly leverage Property 1, it can reduce $D_a$ to a small number *i.e.,* $0.5e$ at a constant cost. Compared to the cross-update strategy that leverages Property 2, it can not only reduce the degree of accumulated error significantly more but also halve the training cost.

### 3.2 SEMANTIC LOSS DECOMPOSITION

**Motivation.** According to memorization effects, neural networks prioritize learning simpler patterns from data (Zhang et al., 2021). The previously followed small-loss criterion leverages this effect: clean labels are learned first by the network, hence exhibiting smaller losses compared to noisy ones. However, the relative magnitude of losses is determined by comparison with other samples *e.g.,* rank sample losses with Top-k algorithm (Han et al., 2018; Jiang et al., 2018) or modeling loss distribution (Li et al., 2020; Permuter et al., 2006). To avoid such costly overheads, here comes a pure idea to leverage memorization effects: In an encoding space where a single loss can be decomposed, the flipped labels share some identical components with the original labels. For flipped labels, the clean components will be learned first and incur smaller losses, while the noisy components result in larger losses. This property can be utilized to identify whether a label contains noise.

Therefore, we design a lightweight plugin to create a detailed distribution in a single loss. The plugin is structurally composed of a pre-prepared non-orthogonal codebook and an auxiliary head at the last layer of the network. It spans an auxiliary space where a loss can be decomposed semantically. Specifically, the codebook transforms the label into hash codes while the head map outputs into feature embedding, respectively. We will first detail the codebook and auxiliary head respectively, and then introduce the selection operation simplified by leveraging both.

**Codebook.** Inspired by Yang et al. (2015), we utilize the favorable properties of Hadamard matrices to construct mappings for category encoding. A $K$-bit Hadamard matrix can generate $2K$ codewords, each $K$ bits long, with a minimum Hamming distance of $\frac{K}{2}$. For $K$-bit hash codes, we construct a $K \times K$ Hadamard matrix. From this, we select $c$ row vectors as category encodings, each with a Hamming distance of $\frac{K}{2}$. Noisy labels $\tilde{y}$ are mapped into hash codes $\tilde{y}'$ through this codebook. Given a classification task with $C$ classes, the mapping is formalized as:

$$H : \tilde{y}_i \in \{0, \cdots, C-1\} \rightarrow \tilde{y}_i' \in \{-1, 1\}^K.$$

**Auxiliary Head.** The auxiliary detection head is an additional three-layer MLP with a Tanh activation function. It shares the same feature extractor with the original classification head and maps the outputs $x$ of neural networks to K-bit feature embeddings $z$, as represented by the function

$$f : x_i \in \mathbb{R}^n \rightarrow z_i \in \mathbb{R}^K.$$

For an ideally clean sample, the distance $\mathbf{d}(\mathbf{z}_i^{(t)}, \mathbf{y}_i)$ between the output $\mathbf{z}_t$ of an ideally clean sample indexed $i$ at time $t$ and its label $\mathbf{y}$ can be expressed as

$$\lim_{t \to \infty} \mathbf{d}(\mathbf{z}_i^{(t)}, \mathbf{y}_i) = \mathbf{0}. \tag{1}$$

The notation $\mathbf{0}$ denotes a vector of zeros, indicating that the distances across different bits uniformly converge towards zero as $t$ approaches infinity.

**Selection Operation.** In this framework, we employ binary cross-entropy (BCE) (Ruby and Yendapalli, 2020) to define the distance $\mathbf{d}$ between predictions and the labels of the network, which indicates whether the network adequately captures the semantics of each component, which is formulated as

$$\mathbf{d}(\mathbf{z}_i, \tilde{\mathbf{y}}_i') = - \left[ \tilde{\mathbf{y}}_i' \odot \log(\mathbf{z}_i) + (1 - \tilde{\mathbf{y}}_i') \odot \log(1 - \mathbf{z}_i) \right]. \tag{2}$$

We apply the arithmetic mean for supervising this head, where the loss function can be articulated as

$$\mathcal{L}_i^{BCE} = -\frac{1}{K} \sum_{j=1}^{K} \left[ \tilde{y}_{ij}' \log(z_{ij}) + (1 - \tilde{y}_{ij}') \log(1 - z_{ij}) \right].$$ (3)

The distance vector $\mathbf{d}(\mathbf{z}_i, \tilde{\mathbf{y}}_i')$ obtained through semantic decomposition directly describes the distribution of sample loss in semantic space. Compared to a single loss value, it provides richer information. To leverage the memorization effect, we use variance to characterize the disparity in the learning degree of the network across different semantic components.

$$\mathrm{Var}(\mathbf{d}(\mathbf{z}_i, \tilde{\mathbf{y}}_i')) = \frac{1}{K} \left( \mathbf{d}(\mathbf{z}_i, \tilde{\mathbf{y}}_i') - \mathcal{L}_i^{BCE} \mathbf{1} \right)^T \left( \mathbf{d}(\mathbf{z}_i, \tilde{\mathbf{y}}_i') - \mathcal{L}_i^{BCE} \mathbf{1} \right).$$ (4)

We set a fixed threshold to distinguish clean samples from noisy samples, which is independent of different samples and different training phases. The identifier $\mathcal{I}_{\mathrm{detection}}$ is updated by detection head as follows:

$$\mathcal{I}_{\mathrm{detection}} = \begin{cases} \mathrm{True} & \text{if } \mathrm{Var}(\mathbf{d}(\mathbf{z}_i, \tilde{\mathbf{y}}_i')) \leq \tau, \\ \mathrm{False} & \text{otherwise.} \end{cases}$$ (5)

Since this threshold is expected to approach zero infinitely, we set it to 0.001.

### 3.3 TRAINING PIPELINE

---
**Algorithm 1** Jump-teaching

---
**Input:** noisy training data $\mathcal{D}$, network parameters $w_f$, learning rate $\eta$, maximum epochs $T_{\max}$, the number of samples $S_{\max}$, identifier table $\mathcal{I}$, clean flag $\phi$
**Output:** $w_f$
1: Initialize $\mathcal{I}[i] \leftarrow$ True for all $i$
2: **for** $t = 1$ to $T_{\max}$ **do**
3:     Permute $\mathcal{D}$ randomly
4:     **for** $s = 1$ to $S_{\max}$ **do**
5:         $\phi \leftarrow \mathcal{I}[s]$
6:         Fetch current sample $\mathcal{D}[s]$
7:         Update $\mathcal{I}[s]$ based on Eq. 8
8:         **if** $\phi$ **then**
9:             Update: $w_f \leftarrow w_f - \eta \nabla(\mathcal{L}^{BCE}(w_f, \mathcal{D}[s]) + \mathcal{L}^{CE}(w_f, \mathcal{D}[s]))$
10:         **end if**
11:     **end for**
12: **end for**
13: **return** $w_f$

---

In this section, we discuss how the proposed plugin can assist in selection. During training, the detection head is on top of the existing network, trained by Eq. 3. Meanwhile, the classification head continues to train normally and is used for inference. However, the two heads exhibit different convergence rates, which leaves room for optimization. The cross-entropy loss, the objective function for classification tasks, is more readily optimizable and thus converges significantly faster than the detection head and can lead to premature over-fitting of noise, resulting in error accumulation. To balance the convergence rate of the two, we employ temperature scaling to calibrate the label probabilities (Guo et al., 2017) $p$ of the classification head. Thus, the soften softmax function will be:

$$\sigma_{\mathrm{softmax}}(p_{ij}) = \frac{\exp(p_{ij}/T)}{\sum_{j=1} \exp(p_{ij}/T)}.$$ (6)

where $T$ is a temperature scaling factor, controlling the convergence rates of the heads. More details are presented in Appendix A.7.

Building on the work of Xiao et al. (2023), we follow the widely accepted principle that assuming the model is well-trained, predictions of clean samples should align with true labels. Based on this

principle, we also make use of the classifier head to apply a straightforward criterion, which further recovers the discarded clean labels. The clean table of the classifier head is then evaluated as:

$$\mathcal{I}_{\text{classifier}} = (\hat{y}_i == \tilde{y}_i). \tag{7}$$

where $\hat{y}_i = \arg\max_j p_i^j$ is the prediction label and $p_i^j$ represents the probability of the $j$-th class for the $i$-th sample. $\tilde{y}_i$ denotes the label of the $i$-th sample. Finally, we can update the table by combining Eq. 5 and Eq. 7:

$$\mathcal{I}' = \mathcal{I}_{\text{detection}} \vee \mathcal{I}_{\text{classifier}}. \tag{8}$$

The algorithm of Jump-teaching is shown in Algorithm 1.

## 4 EXPERIMENTS

In this section, we demonstrate the effectiveness of our proposed method, Jump-teaching, compared with the state-of-the-art in Sec. 4.1. The ablation study is illustrated in Appendix A.8. As the jump-update strategy and semantic loss decomposition are the two orthogonal components of this method, we thoroughly examine each of them in Sec. 4.2 and Appendix A.9. The details of selected samples are described in Appendix A.10.

**Noisy Benchmark Datasets.** We verify the experiments on three benchmark datasets, including *CIFAR-10*, *CIFAR-100* and *Clothing1M* (Xiao et al., 2015). These datasets are popular for evaluating noisy labels. They are summarized in Appendix A.11. Following the setup on (Li et al., 2020; Liu et al., 2020), we simulate two types of label noise: symmetric noise, where a certain proportion of labels are uniformly flipped across all classes, and asymmetric noise, where labels are flipped to specific classes, *e.g.*, $bird \rightarrow airplane$, $cat \leftrightarrow dog$. Assume $\epsilon$ denotes the noise ratio, their mathematical definitions are in Appendix A.12. We also experiment on instance-dependent noise (IDN) and pairflip-45 noise, results are reported in Appendix A.13.

**Baselines.** To be more convincing, we compare the competitive methods of LNL. These methods are as follows: Standard, which is simply the standard deep network trained on noisy datasets, Decoupling (Malach and Shalev-Shwartz, 2017), Co-teaching (Han et al., 2018), Co-teaching+ (Yu et al., 2019), PENCIL (Yi and Wu, 2019), TopoFilter (Wu et al., 2020), ELR (Liu et al., 2020), FINE (Kim et al., 2021), SPRL (Shi et al., 2023), RML (Li et al., 2024), APL (Ma et al., 2020), CDR (Xia et al., 2021), MentorNet (Jiang et al., 2018), SIGUA (Han et al., 2020), JoCoR (Wei et al., 2020) and CoDis (Xia et al., 2023). A brief overview of the available source code of these methods is illustrated in Appendix A.14.

**Experimental Settings.** All experiments operate on a server equipped with an NVIDIA A800 GPU and PyTorch platform. In the following experiments, Jump-teaching *almost* employs the same configuration. It trains the network for 200 epochs by SGD with a momentum of 0.9, a weight decay of $1e-3$, and a batch size of 128. The initial learning rate is set to 0.2, and a cosine annealing scheduler finally decreases the rate to $5e-4$. The warm-up strategy is utilized by Jump-teaching, and the warm-up period is 30 epochs. After the warm-up period, we augmented the data as detailed in the Appendix A.15. The threshold of variance $\tau = 0.001$ is discussed in Appendix A.16. The jump step is stated in Appendix A.17. Exceptionally, we set the weight decay as $5e-4$ to facilitate learning on fewer available samples when the noise ratio $\epsilon$ equals 50% and 80% in *CIFAR-100*, respectively. The single network of Jump-teaching employs three types of backbone networks to fulfill different requirements of the experimental design, such as PreActResNet-18 (He et al., 2016a), ResNet-18, and three layers of neural network. Moreover, the architectures of these backbones and the auxiliary head are illustrated in Appendix A.18. The baseline methods fully follow the experimental setup in the literature (Han et al., 2018; Li et al., 2020).

### 4.1 COMPARISON WITH THE STATE-OF-THE-ARTS

**Synthetic Noisy Benchmark.** We compare our proposed method with the following representative approaches: Standard, Decoupling, Co-teaching, Co-teaching+, PENCIL, TopoFilter, ELR, FINE, and SPRL. As shown in Table 1, Jump-teaching demonstrates superior performance with different noise settings. With the symmetric noise ratio $\epsilon = 0.8$, its accuracy has improved by 13.3% and 16.2% on two datasets. This indicates its strong robustness.

Table 1: Test accuracy(%) on *CIFAR-10* and *CIFAR-100* with symmetric and asymmetric noise. All methods employ PreActResNet-18 and train 200 epochs with three trials. The best results are highlighted in bold.

| Dataset | CIFAR-10 | | | | CIFAR-100 | | | |
|---|---|---|---|---|---|---|---|---|
| Noise Type | Sym. | | | Asym. | Sym. | | | Asym. |
| Noise ratio | 0.2 | 0.5 | 0.8 | 0.4 | 0.2 | 0.5 | 0.8 | 0.4 |
| Standard | 84.6±0.1 | 62.4±0.3 | 27.3±0.3 | 75.9±0.4 | 56.1±0.1 | 33.6±0.2 | 8.2±0.1 | 40.1±0.2 |
| Decoupling('17) | 86.4±0.1 | 72.9±0.2 | 48.4±0.6 | 83.3±0.2 | 53.3±0.1 | 28.0±0.1 | 7.9±0.1 | 39.9±0.4 |
| Co-teaching('18) | 89.9±0.6 | 67.3±4.2 | 28.1±2.0 | 79.2±0.5 | 61.8±0.4 | 34.7±0.5 | 7.5±0.5 | 40.0±1.2 |
| Co-teaching+('19) | 88.1±0.0 | 61.8±0.2 | 22.3±0.6 | 58.2±0.2 | 54.5±0.1 | 27.6±0.1 | 8.4±0.1 | 19.9±0.3 |
| PENCIL('19) | 88.2±0.6 | 73.4±1.5 | 36.0±0.7 | 77.3±3.4 | 57.4±1.0 | 11.4±3.2 | 5.4±1.2 | 45.7±0.9 |
| Topofilter('20) | 89.5±0.1 | 84.6±0.2 | 45.9±2.6 | 89.9±0.1 | 63.9±0.8 | 51.9±1.7 | 16.9±0.3 | 66.6±0.7 |
| FINE('21) | 90.2±0.1 | 85.8±0.8 | 70.8±1.8 | 87.8±0.1 | 70.1±0.3 | 57.9±1.2 | 22.2±0.7 | 53.5±0.8 |
| SPRL('23) | 91.7±0.2 | 88.4±0.6 | 63.9±1.4 | 89.8±0.5 | 69.5±0.8 | 57.2±1.6 | 23.8±2.2 | 58.7±1.3 |
| RML('24) | 92.2±0.1 | 88.3±0.3 | 35.3±1.7 | 79.8±4.0 | 67.2±0.2 | 62.0±0.5 | 15.7±0.8 | 64.5±0.6 |
| Jump-teaching | **94.8±0.1** | **92.2±0.1** | **84.1±1.1** | **90.7±0.3** | **72.7±0.5** | **67.1±0.2** | **40.0±1.1** | **68.4±0.7** |

**Efficiency Analysis.** We compare the efficiency of Jump-teaching with the above representative methods and follow the same settings as the synthetic noisy benchmark. The efficiency of these methods is evaluated by throughput and peak memory usage. Throughput is the measurement of the rate at which a method processes picture frames, expressed in *thousands* of frames per second (kfps). Peak memory usage refers to the maximum amount of memory consumed by a method during its execution. We observe the efficient metrics of these methods with symmetric noise ratio $\epsilon = 0.5$ in Table 2, while the accuracy of these methods is illustrated in Table 1. As shown in Table 1 and Table 2, our method achieves almost up to $2.5\times$ speedup, $0.46\times$ peak memory footprint, and superior robustness over all methods. Thus, Jump-teaching achieves optimal empirical results in terms of training speed, memory usage, and accuracy.

Table 2: Test computational and storage efficiencies.

| | Standard | Decoupling | PENCIL | Co-teaching | Co-teaching+ | Topofilter | FINE | SPRL | Jump-teaching |
|---|---|---|---|---|---|---|---|---|---|
| Single network | ✓ | ✗ | ✓ | ✗ | ✗ | ✓ | ✓ | ✓ | ✓ |
| Throughput (kfps) ↑ | 4.16 | 1.72 | 3.13 | 1.81 | 2.63 | 1.61 | 2.17 | 2.94 | **4.07** |
| Peak mem (GB) ↓ | 0.68 | 1.50 | 0.73 | 1.53 | 1.53 | 1.40 | 0.94 | 0.77 | **0.70** |

**Real-World Noisy Benchmark.** We operate experiments with three trials and report mean value on the *Clothing1M* dataset. Our proposed method is compared with the following representative approaches: APL, CDR, MentorNet, Decoupling, Co-teaching, Co-teaching+, JoCoR, and CoDis. Jump-teaching employs ResNet-50 as the backbone, which is pre-trained on *ImageNet* (Deng et al., 2009) dataset and follows the same training setup in Xia et al. (2023). As shown in Table 3, the robustness of Jump-teaching for real-world noisy labels is favorable when compared to other methods.

Table 3: Test accuracy(%) on *Clothing1M*.

| Method | APL | CDR | MentorNet | SIGUA | Co-teaching | Decoupling | Co-teaching+ | JoCoR | CoDis | Jump-teaching |
|---|---|---|---|---|---|---|---|---|---|---|
| Accuracy | 54.46 | 66.59 | 67.25 | 65.37 | 67.94 | 67.65 | 63.83 | 69.06 | 71.60 | **71.93** |

## 4.2 EXPERIMENTS ON JUMP-UPDATE STRATEGY

The jump-update strategy is not only suitable for Jump-teaching but also easily integrated into other methods in LNL. Co-teaching (Han et al., 2018) and DivideMix (Li et al., 2020) can be regarded as the candidates in supervised-only and semi-supervised methodologies of LNL. We collaborate the jump-update strategy with these two methods respectively. The results of experiments on Clothing1M are reported in Appendix A.19.

**Collaboration with Supervised-Only Approaches.** We utilize ResNet-18 as our backbone architecture. We use only one network from co-teaching and replace cross-update strategy with jump-update strategy, termed J-co-teaching. As Table 4 shows, J-Co-teaching achieves comparable results to

Co-teaching with all noise conditions and significantly outperforms Co-teaching in extreme noise scenarios such as $\epsilon = 0.8$ and $\epsilon = 0.9$. In the case of using a single network, J-Co-teaching significantly outperforms Co-teaching under all noise conditions.

Table 4: Comparison of test accuracies(%) for Co-teaching and J-Co-teaching on *CIFAR-10* and *CIFAR-100* with different noise types and ratios.

| Dataset | | *CIFAR-10* | | | | | *CIFAR-100* | | | |
|---|---|---|---|---|---|---|---|---|---|---|
| Noise type | | Sym. | | | Asym. | | Sym. | | | Asym. |
| Methods/Noise ratio | 0.2 | 0.5 | 0.8 | 0.9 | 0.4 | 0.2 | 0.5 | 0.8 | 0.9 | 0.4 |
| Co-teaching (2*ResNet18) | 91.23 | **87.94** | 47.69 | 18.40 | 85.24 | 66.22 | **56.25** | 20.08 | 3.82 | **43.91** |
| Co-teaching (1*ResNet18) | 88.55 | 84.39 | 43.21 | 15.88 | 78.27 | 61.61 | 50.97 | 16.58 | 3.61 | 39.63 |
| J-Co-teaching (1*ResNet18) | 91.52 | 87.82 | **74.24** | **38.41** | 85.99 | 66.31 | 55.39 | **26.55** | 9.16 | 43.53 |

**Collaboration with Semi-Supervised Approaches.** DivideMix not only uses two networks to exchange error flow but also employs them to collaboratively create pseudo-labels. Consequently, we establish three baselines following Li et al. (2020): DivideMix, DivideMix with $\theta_1$ test, and DivideMix w/o co-training. DivideMix with $\theta_1$ test utilizes a single network for sample selection, while DivideMix w/o co-training retains a single sample and employs self-update for updating. In each case, we replace the original update strategy with the jump-update strategy. As Tabel 5 shows, the jump-update strategy significantly improves the accuracy of all baselines, particularly under extreme noise ratios. This indicates that it can effectively overcome selection bias without incurring additional costs.

Table 5: Comparison of test accuracies(%) for DivideMix and J-DivideMix on *CIFAR-10* and *CIFAR-100* with different noise. The best and the mean value of the last ten epochs of accuracy is reported.

| Dataset | | CIFAR-10 | | | | | CIFAR-100 | | | |
|---|---|---|---|---|---|---|---|---|---|---|
| Noise type | | Sym. | | | Asym. | | Sym. | | |
| Methods/Noise ratio | | 0.2 | 0.5 | 0.8 | 0.9 | 0.4 | 0.2 | 0.5 | 0.8 | 0.9 |
| DivideMix | Best | 96.1 | 94.6 | 93.2 | 76.0 | **93.4** | 77.3 | 74.6 | 60.2 | 31.5 |
| | Last | 95.7 | 94.4 | 92.9 | 75.4 | **92.1** | 76.9 | **74.2** | **59.6** | 31.0 |
| J-DivideMix | Best | **96.3** | **94.9** | **93.5** | **78.5** | 93.0 | **77.7** | **74.7** | 60.0 | **32.3** |
| | Last | **96.0** | **94.6** | **93.2** | **77.7** | 91.9 | **77.3** | 74.2 | 59.7 | **32.5** |
| DivideMix with $\theta_1$ test | Best | 95.2 | 94.2 | **93.0** | 75.5 | 92.7 | 75.2 | 72.8 | 58.3 | 29.9 |
| | Last | 95.0 | 93.7 | 92.4 | 74.2 | 91.4 | 74.8 | 72.1 | 57.6 | 29.2 |
| J-DivideMix with $\theta_1$ test | Best | **96.2** | **95.0** | **93.0** | **80.7** | **93.3** | **77.7** | **73.6** | **58.5** | **31.4** |
| | Last | **95.8** | **94.8** | 92.7 | 79.4 | 92.2 | 77.3 | 73.2 | 57.8 | 30.8 |
| DivideMix w/o co-training | Best | 95.0 | 94.0 | 92.6 | 74.3 | 91.9 | 74.8 | 72.3 | **56.7** | 27.7 |
| | Last | 94.8 | 93.3 | 92.2 | 73.2 | 90.6 | 74.1 | 71.7 | **56.3** | 27.2 |
| J-DivideMix w/o co-training | Best | **95.2** | **94.5** | **93.0** | **80.7** | **92.4** | **75.0** | **72.4** | 56.3 | **27.8** |
| | Last | 94.7 | **94.0** | 92.3 | 79.3 | 91.3 | 74.2 | 71.7 | 55.3 | **27.6** |

## 5 CONCLUSION

We propose Jump-teaching, an ultra robust and efficient framework to combat label noise. Our work effectively mitigates the sample-selection bias in a single network which enables our approach to demonstrate outstanding performance across a broad array of experiments, particularly in extreme noise conditions. Moreover, with decomposed semantic information of sample losses for selection, Jump-teaching overcomes the limitations of the small-loss criterion and achieves more effective selection. Extensive experimental validation confirms that our method surpasses current state-of-the-art techniques in both robustness and efficiency. In the future, we hope to explore how the fluctuations of parameters influence the selection capability of the network. We did not employ popular semi-supervised techniques to further exploit the identified noisy samples, however, we believe that our method can be further extended by these techniques.

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
