# A    Supplemental material

## A.1    Discussion about Disagreement

We test a three-layer MLP on *CIFAR-10* with symmetric noise ratio $\epsilon = 0.5$ and $\epsilon = 0.8$, and quantify the disagreement for three update strategies. *Intersection over Union* (IoU) between two selections are utilized for the quantitative evaluation, while the update strategies are the Cross-update, Self-update, and Jump-update. Specifically, Intersection computes the number of commonly selected samples in two selections, while Union calculates the total number of selected samples in two selections. In the cross-update strategy, the disagreement emerges in a different network, while in the other two strategies, it emerges during different epochs.

As shown in Fig. 2(a) and Fig. 2(b), the single network in Self-Update exhibits significantly greater disagreement between epochs than the dual-network setup in Cross-update, which suggests a huge potential for mitigating bias. However, self-update does not leverage this advantage, resulting in the worst performance among strategies. In contrast, jump-update strategy, which leverages previous selections for model updates, displays slightly lower disagreement compared to self-update while achieving remarkable performance. Furthermore, we observe that the error band of Jump-Update is smaller than the other two methods, indicating more stable network updates. This reflects the robustness of the network against noise to some extent. These results are shown in Fig. 2(c) and Fig. 2(d): Jump-Update, with the smallest error band, achieves the best performance, while self-update, with the largest error band, performs the worst.

## A.2    Discussions about "Disagreement"

In this section, we discuss the concept of "disagreement" and its relationship with prior related works (Wei et al., 2022; Yuan et al., 2023). The "disagreement" discussed in our paper differs fundamentally from similar concepts in previous studies in terms of its definition, described objects, application, calculation, and focus. Unlike the "fluctuation" defined in (Wei et al., 2022), which refers to a sample being correctly classified at one step but misclassified in the next, or the "First-time k-epoch Learning (FkL)" introduced in (Yuan et al., 2023), which measures the first training epoch where an instance is consistently predicted to its given label for k consecutive epochs, our "disagreement" is derived from (Han et al., 2018) and refers to differences in selection behaviors within networks. These differences are not tied to individual samples or labels but rather describe variations in network behaviors over iterations.

The objects described in these concepts are also distinct. While "fluctuation" and "FkL" capture variations in the predictions of individual samples, "disagreement" focuses on the characteristics of networks. For instance, "disagreement" in our work examines the behavior of a single network across different iterations, whereas in Han et al. (2018), "disagreement" describes an attribute of two networks.

The applications of these concepts further emphasize their differences. As a network-level characteristic, "disagreement" is employed to design strategies for updating models. In contrast, "fluctuation" and "FkL," being sample-level characteristics, are primarily used to define selection criteria for determining whether a sample should be included in the training process. Additionally, the methods for calculating and interpreting these concepts vary significantly. Both "fluctuation" and "FkL" emphasize consistency between model predictions and true labels, aiming to identify clean samples. On the other hand, "disagreement" is independent of labels and is quantified using the Intersection Over Union (IoU) metric between selected sets, enabling it to address selection bias in model updates.

Our work contributes novel insights into the concept of "disagreement." We are the first to demonstrate that disagreement within a single network not only exists but also persists during training, as illustrated in Fig. 2(a) and Fig. 2(b). Additionally, we find that single-network disagreement is even more significant than that observed between two networks, a discovery that underscores its importance in understanding network behavior. To further solidify this understanding, we introduce the IoU metric to quantify and visualize disagreement, providing a new lens through which to analyze and mitigate selection bias in model updates.

## A.3 EXAMPLE OF JUMP-UPDATE STRATEGY

We would like to give a simple example to illustrate the Jump-update Strategy for quick understanding. Suppose we have three iterations, iter0, iter1, and iter2. After the parameters of iter1 are updated through backpropagation, the resulting updated model is referred to as iter2. When training in iter2, the model only used the clean samples selected by iter0, which creates a 'disagreement' due to the inconsistency between the iteration where sample selection occurs (iter0) and the iteration where the model is updated (iter1). The behavior of the update exhibits a jump form. In this example, the current iteration $i$ equals 2, and jump-step $S$ equals 2. Iter0 is the ancestor iteration of iter2.

## A.4 PROOF OF PROPERTY 1

The key of proving $D_A \propto n$ is to prove $N_A = N_{iteration}$. Under the hypothesis that the error flow is an uninterrupted model, the situation can be categorized into two types: (1) errors accumulate in a single network; (2) errors accumulate across multiple networks. Through the analysis of the following two types, Property 1 is proved.

**Errors Accumulate in A Single Network.** When errors accumulate in a single network, one accumulation is generated from one sample selection and one model update. As is obvious that the single network operates only one sample selection and a parameter updating in each iteration. Therefore, $N_A = N_{iteration}$.

**Errors Accumulate across Multiple Networks.** When errors accumulate across multiple networks, errors first pass through each network once before being transmitted back to the original network. We denote the number of networks by $N_{net}$. Error accumulation can only occur when the updates of the network are guided by its own selections. Thus, this process requires $N_{net}$ sample selections and $N_{net}$ parameter updates. Similarly, in each iteration, all networks collectively undergo $N_{net}$ sample selections and $N_{net}$ parameter updates.

## A.5 DISCUSSION ON ERROR FLOW

First of all, let us review the formulation $N_a = \frac{N_A}{N_f}$, mentioned in Property 2. The following discussion is helpful for the understanding of this formulation. We denote the number of samples as $n_s$ and the number of mini-batches as $n_b$. If the network updates the identifier table after updating the model when the data for selection is a single sample, $N_f = n_s$. When it comes to mini-batches, $N_f = n_b$. In all of these conditions, $N_a = e$. Thus the single network can be enhanced by integrating the jump-update strategy. If the network updates the identifier table before updating parameters, $N_f$ will be doubled so that $N_a = 0.5e$. The cross-Update strategy uses two networks to mitigate bias, it changed $N_f$ to 2. This shows that the cross-update strategy can be incorporated into our framework and does not require the use of two networks as well as the maintenance of two training processes. Moreover, we provide two methods to reduce $N_a$: the most direct method is to control the update frequency of the identifier table, or by adding new identifier tables to maintain the updates per iteration, each doubling of the identifier table reduces $N_a$ by half. This allows us to control $N_a$ flexibly. Empirical evidence has shown that maintaining $N_a$ in the hundreds is reasonable. Limiting $N_a$ to a single-digit number can lead to the model updating too slowly and over-fitting noise.

## A.6 DISCCUSION ON INTIAL BIAS

Initial selection bias and number of accumulations, *i.e.,* $N_a$, will jointly affect the overall degree of accumulated error $D_A$. Initial Bias is manifested in the form of initial identifier tables which can be obtained in three ways: the first is randomly selection, the second involves updates during warm-up, and the third approach involves labeling before training, which is the usual practice. For ease of subsequent discussion, we call them Random initiation, Warm-up initiation, and Instant initiation. To analyze the impact of initial bias on error accumulation, we observe the subsequent performance of the network by controlling the initial bias, using a single network as the baseline. The update frequency of identifier table *i.e.,* $r$ is set at 10%, and three different initiations are employed. The first two tables are unbiased as they are not affected by selection, whereas Instant initiation is biased as it acquires tables during the second epoch of training and is affected by an increasingly deepening bias. Experimental results are shown in Fig. 5. The identifier table from Random initiation is less accurate

than that obtained through Instant initiation, yet its performance is significantly better, demonstrating the effectiveness of avoiding bias. Warm-up initiation gets the best result, as it avoids performance degradation caused by the high error rate of the initial table.

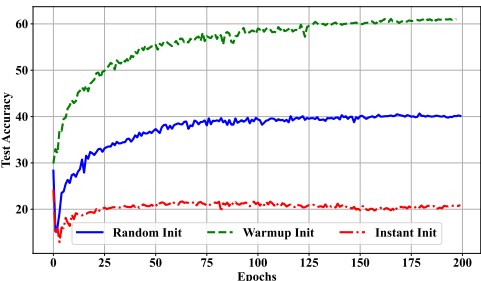

Figure 5: Effect of initial bias on training.

## A.7 LOSS CONVERGENCE

To show how the different values of scaler $T$ affect the convergence rates, we present the results of four different temperature values in Fig. 6. When $T$=1, the Binary Cross-Entropy (BCE) loss fails to converge properly. The cases for $T = 2$, 3, and 4 illustrate the effects of temperature scaling. Specifically, temperature scaling slows down the convergence speed of the classification head, which can be obviously observed during the warm-up phase, while it accelerates the convergence of the detection head, which can be obviously observed during the training phase. Notably, as the temperature T increases, the convergence rate of the Cross-Entropy (CE) loss decreases.

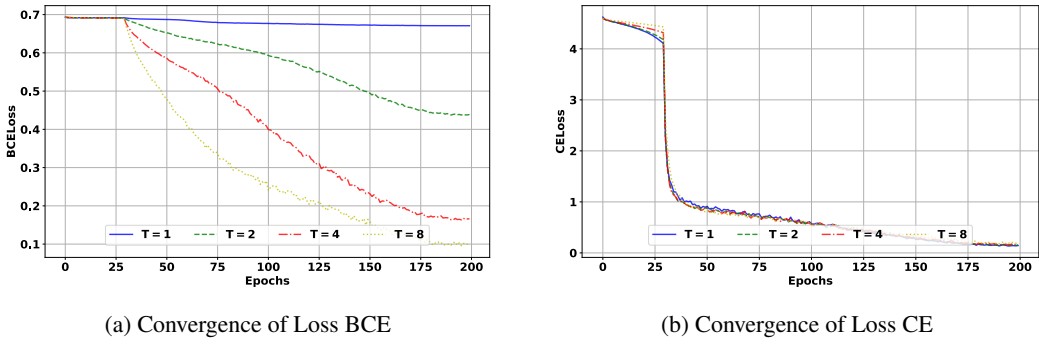

(a) Convergence of Loss BCE    (b) Convergence of Loss CE

Figure 6: Convergence of losses with different temperature scaling factors $T$ under CIFAR-100 Sym. $\epsilon = 0.8$.

## A.8 ABLATION STUDY

We evaluate three components of Jump-teaching: data augmentation (AUG), jump-update strategy (JU), and sample selection (SS) with an ablation study. Results are shown in Table 6.

**Effectiveness of Data Augmentation.** When only a limited number of samples are available, such as in the presence of symmetric noise ratio $\epsilon = 0.8$, data augmentation enables the network to gain knowledge of the intrinsic characteristics of the images themselves. Therefore, it can significantly enhance the learning capabilities of the networks for these samples.

**Effectiveness of Jump-update Strategy.** Jump-update strategy can effectively overcome error accumulation, allowing it to perform excellently with extreme noise. Without the jump-update strategy, the network is almost unable to learn on *CIFAR-10* with the symmetric noise ratio $\epsilon = 0.8$.

**Effectiveness of Sample Selection.** If noisy samples are not filtered out at all, the network can easily fit them, resulting in poor generalization performance. This has been verified among diverse noise settings.

Table 6: Ablation Study.

| Modules | | | CIFAR-10 | | | | CIFAR-100 | | | |
|---|---|---|---|---|---|---|---|---|---|---|
| AUG | JU | SS | Sym. 20% | Sym. 50% | Sym. 80% | Asym. 40% | Sym. 20% | Sym. 50% | Sym. 80% | Asym. 40% |
| × | ✓ | ✓ | 94.12 | 89.59 | 74.94 | 85.34 | **74.69** | 65.87 | 32.66 | 67.30 |
| ✓ | × | ✓ | 88.33 | 81.27 | 10.01 | 56.28 | 68.32 | 65.27 | 28.71 | 50.40 |
| ✓ | ✓ | × | 85.42 | 63.21 | 30.18 | 78.19 | 61.88 | 39.35 | 12.90 | 45.17 |
| × | × | ✓ | 92.21 | 87.24 | 9.99 | 57.56 | 74.19 | 65.98 | 29.23 | 57.05 |
| ✓ | ✓ | ✓ | **94.77** | **92.11** | **82.81** | **90.91** | 72.15 | **66.91** | **40.81** | **67.66** |

A.9 THE SIZE OF HADAMARD CODEBOOK

As demonstrated in Table 7, we find that the performance of the network does not exhibit a clear correlation with the code length. This indicates that our approach differs from traditional deep hashing methods (Yang et al., 2015; Yuan et al., 2020; Jose et al., 2022) which rely on learning the semantic features of samples. It is common sense that the performance of such methods is closely associated with the dimensionality of the high-dimensional space, as more dimensions typically provide richer information. However, our method only employs non-mutual exclusion of encoding, not high-dimensional properties, to reflect patterns of memorization effects. Therefore, our approach is robust to code length.

Table 7: Test accuracy(%) on CIFAR-10 and CIFAR-100 with symmetric and asymmetric noise.

| Dataset | CIFAR-10 | | | | CIFAR-100 | | | |
|---|---|---|---|---|---|---|---|---|
| Noise type | Sym. | | | Asym. | Sym. | | | Asym. |
| Code length/Noise ratio | 0.2 | 0.5 | 0.8 | 0.4 | 0.2 | 0.5 | 0.8 | 0.4 |
| 8 | 94.66 | 91.85 | 82.75 | 90.56 | - | - | - | - |
| 16 | 94.7 | 92.00 | 82.87 | 90.79 | - | - | - | - |
| 32 | 94.74 | 92.3 | 83.28 | 90.99 | - | - | - | - |
| 64 | 94.67 | 92.10 | 83.41 | 90.51 | 72.14 | 66.91 | 40.81 | 67.66 |
| 128 | 94.62 | 91.95 | 83.1 | 90.55 | 72.04 | 66.39 | 38.25 | 68.86 |

A.10 DETAILS OF SAMPLE SELECION

We record the number and ratio of clean samples selected in each epoch. In moderate noise, this data is reported for noise condition Sym. $\epsilon = 0.2$, Sym. $\epsilon = 0.5$, Sym. $\epsilon = 0.8$ in the CIFAR-10 and CIFAR-100 datasets and is presented in Fig. 7. In extreme noise, we report the results in Sym. $\epsilon = 0.9$ in the CIFAR-10 and is presented in Fig. 8.

**Moderate Noise.** It can be observed that the number of clean samples continuously increases and stabilizes during the last 20 epochs, presented in Fig. 7(a) and Fig. 7(c). As for Fig. 7(b) and Fig. 7(d), the ratio of clean samples increases steadily during the early and middle stages of training, indicating that the selection ability of the network improves as its performance increases. However, in the later stages (epochs range from 150 to 200), the ratio of clean samples begins to decline, primarily due to the model starting to fit the noisy data, which results in more noisy samples being misclassified as clean.

**Extreme Noise.** As illustrated in Fig. 8(b), methods utilizing the jump-update strategy achieve a higher ratio in clean samples, demonstrating the effectiveness of the jump-update strategy in mitigating selection bias. Furthermore, Fig. 8(a) highlights that Jump-teaching significantly outperforms others in the number of selected clean samples, underscoring the effectiveness of Semantic Loss Decomposition in detecting noisy samples.

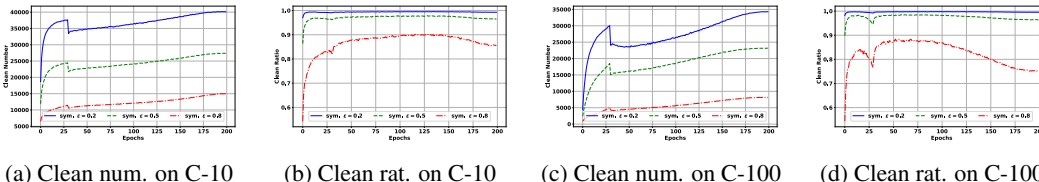

(a) Clean num. on C-10     (b) Clean rat. on C-10     (c) Clean num. on C-100     (d) Clean rat. on C-100

Figure 7: Clean number (num.) and ratio (rat.) of selected samples on CIFAR-10 (C-10) and CIFAR-100 (C-100) with different noise types and ratios.

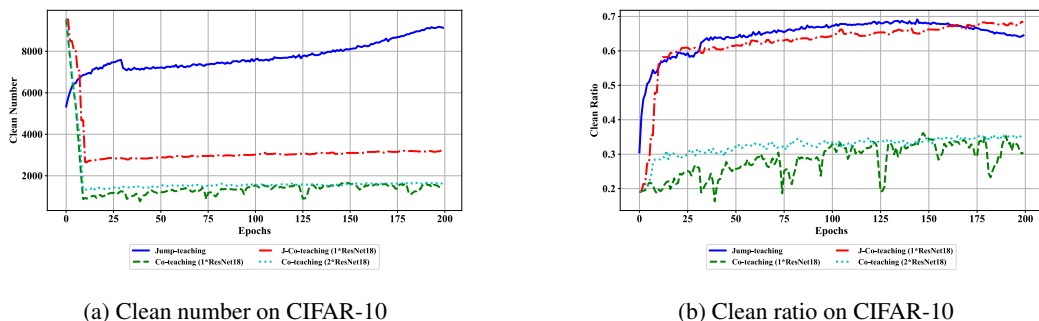

(a) Clean number on CIFAR-10        (b) Clean ratio on CIFAR-10

Figure 8: Clean number and ratio of selected samples on CIFAR-10 and CIFAR-100 with extreme noise.

## A.11 THE SUMMARY OF THE BENCHMARK DATASETS

The summary of three benchmark datasets, *CIFAR-10*, *CIFAR-100* and *Clothing1M* is illustrated in Table 8. Synthetic noisy benchmark in Sec. 4.1 is evaluated on *CIFAR-10*, and *CIFAR-100*, respectively, while real-world noisy benchmark in Sec. 4.1 is evaluated on *Clothing1M*. Specifically, the *CIFAR-10* dataset is a standard dataset widely used for image classification tasks. It consists of $60,000$ color images with a resolution of $32 \times 32$ pixels, divided into 10 categories, each containing $6,000$ images. The dataset is divided into $50,000$ training images and $10,000$ testing images. Similar to the former, the *CIFAR-100* dataset contains more categories. It consists of $60,000$ color images of $32 \times 32$ pixels, divided into 100 categories, with each category containing 600 images. *Clothing1M* dataset is a large-scale image classification dataset specifically designed for clothing recognition. It contains over $1,000,000$ labeled images of clothing, covering 14 different categories such as skirts, shirts, coats, shoes, *etc*. Images in the *Clothing1M* dataset typically do not have a fixed standard resolution, as they are sourced from the internet.

Table 8: Summary of datasets used in the experiments.

| Dataset | # of training | # of testing | # of class | Image size |
|---|---|---|---|---|
| *CIFAR-10* | 50,000 | 10,000 | 10 | 32x32 |
| *CIFAR-100* | 50,000 | 10,000 | 100 | 32x32 |
| *Clothing1M* | 120,000 | 10,000 | 14 | not fixed |

## A.12 THE SIMULATION OF NOISE

As the datasets are clean, we follow Li et al. (2020) and simulate symmetric and asymmetric noise.

**Symmetric Noise.** Symmetric noise assumes that label errors occur randomly and are independent of the true labels. For a classification problem with $C$ classes, symmetric noise is typically defined as in Eq. 9. The formula includes two things: 1) Each correct label remains unchanged with probability $1 - \epsilon + \frac{\epsilon}{C}$. 2) Each correct label $i$ changes to one of the all $C$ classes $j$ with probability $\epsilon$. If the noise is uniformly distributed, the probability of changing to any other class is $\frac{\epsilon}{C}$.

$$\begin{cases} 1 - \epsilon + \frac{\epsilon}{C} & \text{if } i = j, \\ \frac{\epsilon}{C} & \text{otherwise.} \end{cases} \quad (9)$$

**Asymmetric Noise.** Asymmetric noise reflects that label errors are more likely to occur between certain classes, typically based on practical scenarios where some classes are visually similar or commonly confused. The probability model for asymmetric noise typically depends on the specific task and data. A simple model might be that the label for a specific class $i$ is incorrectly marked as class $j$ with probability $\epsilon_{ij}$. The asymmetric noise can be formulated in Eq. 10 as follows.

$$P(\tilde{Y} = j \mid Y = i) = \epsilon_{ij}, \quad (10)$$

where $\epsilon_{ij}$ can be non-zero when $i \neq j$, reflecting specific mislabeling scenarios. where $Y$ represents the true label, $\tilde{Y}$ represents the potentially noisy label, and $\epsilon$ is the noise ratio.

Overall, the symmetric noise assumes that the probability of mislabeling is uniform across all categories, whereas the asymmetric noise considers the similarity between categories. The choice of noise typically depends on the characteristics of the dataset and the specific requirements of the task.

### A.13 MORE EXPERIMENTS ON DIFFERENT NOISE TYPES

Although our experiments tested five types of noise in Sec. 4, the additional experiments of pairflip-45 and instance-dependent label noise (IDN) are illustrated here. Results are shown in Table 9 and Table 10, respectively. On pairflip noise, Jump-teaching also exhibits its strong performance, with an increase of 6.73% and 9.95% on CIFAR-10 and CIFAR-100, respectively. Meanwhile, our algorithm maintains a huge advantage in instance-dependent noise. The leading margins are 3.95% (CIFAR-10 IDN. 0.2), 6.74% (CIFAR-10 IDN. 0.4), 10.2% (CIFAR-10 IDN. 0.6), 4.87% (CIFAR-100 IDN. 0.2), 4.3% (CIFAR-100 IDN. 0.4) and 12.67% (CIFAR-100 IDN. 0.6). This indicates that our algorithm is also extremely robust under IDN settings.

Table 9: Performance of methods on CIFAR-10 and CIFAR-100 with pairflip $\epsilon = 0.45$ noise.

| Methods | CIFAR-10 | CIFAR-100 |
|---|---|---|
| CE | $50.22 \pm 0.43$ | $21.59 \pm 0.87$ |
| Co-teaching | $52.99 \pm 0.16$ | $33.22 \pm 0.66$ |
| Co-teaching+ | $55.19 \pm 0.27$ | $29.26 \pm 0.15$ |
| PENCIL | $54.58 \pm 2.26$ | $24.97 \pm 0.57$ |
| SPRL | $90.54 \pm 0.02$ | $53.62 \pm 1.07$ |
| FINE | $77.09 \pm 0.10$ | $41.62 \pm 1.01$ |
| Topofilter | $84.60 \pm 0.45$ | $52.40 \pm 1.42$ |
| Jump-teaching | $\mathbf{91.33 \pm 0.23}$ | $\mathbf{62.35 \pm 0.74}$ |

Table 10: Performance of methods on CIFAR-10 and CIFAR-100 with varying IDN noise ratios.

| Methods | CIFAR-10 | | | CIFAR-100 | | |
|---|---|---|---|---|---|---|
| | 0.2 | 0.4 | 0.6 | 0.2 | 0.4 | 0.6 |
| CE | $85.45 \pm 0.57$ | $76.23 \pm 1.54$ | $59.75 \pm 1.30$ | $57.79 \pm 1.25$ | $41.15 \pm 0.83$ | $25.68 \pm 1.55$ |
| Co-teaching | $88.87 \pm 0.24$ | $73.00 \pm 1.24$ | $62.51 \pm 1.98$ | $43.30 \pm 0.39$ | $23.21 \pm 0.57$ | $12.58 \pm 0.51$ |
| Co-teaching+ | $89.80 \pm 0.28$ | $73.78 \pm 1.39$ | $59.22 \pm 6.34$ | $41.71 \pm 0.78$ | $24.45 \pm 0.71$ | $12.58 \pm 0.51$ |
| PENCIL | $86.23 \pm 1.23$ | $69.88 \pm 1.39$ | $44.16 \pm 1.58$ | $57.41 \pm 0.69$ | $48.38 \pm 0.52$ | $29.69 \pm 3.03$ |
| FINE | $90.85 \pm 1.21$ | $84.53 \pm 0.77$ | $61.24 \pm 2.02$ | $60.14 \pm 0.015$ | $43.05 \pm 1.43$ | $28.88 \pm 2.01$ |
| Topo | $89.66 \pm 0.86$ | $85.95 \pm 1.02$ | $75.28 \pm 0.64$ | $64.41 \pm 0.37$ | $55.00 \pm 0.82$ | $31.61 \pm 1.45$ |
| SPRL | $87.99 \pm 0.03$ | $86.15 \pm 0.81$ | $75.01 \pm 0.74$ | $67.66 \pm 0.17$ | $64.55 \pm 0.19$ | $47.16 \pm 1.97$ |
| Jump-teaching | $\mathbf{94.80 \pm 0.28}$ | $\mathbf{92.89 \pm 0.33}$ | $\mathbf{85.48 \pm 0.41}$ | $\mathbf{72.53 \pm 0.21}$ | $\mathbf{68.85 \pm 0.27}$ | $\mathbf{59.83 \pm 0.09}$ |

## A.14 A BRIEF OVERVIEW OF COMPARED METHODS

We reimplemented Decoupling, Co-teaching, Co-teaching+, and PENCIL with the publicly available code[*]. Similarly, FINE, SPRL, and TopoFilter were re-implemented by their respective official repositories[†][‡][§]. Results in Table 3 are cited from (Xia et al., 2023). The outcomes for Decoupling, Co-teaching+, and PENCIL on *CIFAR-10* and *CIFAR-100* with symmetric noise, shown in Table 1, are derived from (Li et al., 2023). Additionally, baseline results featured in Table 5 for DivideMix, DivideMix with $\theta_1$ test, and DivideMix without co-training are sourced from (Li et al., 2020).

## A.15 DATA AUGMENTATION

When the noise is severe, there are very few clean samples. Thus, the network struggles to acquire sufficient knowledge from a small number of available samples. Therefore, we adopt a data augmentation strategy following Cubuk et al. (2020) and linearly interpolate the augmented image with its weak view generated by random cropping and horizontal flipping. This enables the network to more easily focus on the key features of the samples. We follow Eq. 11 to combine weak and strong views.

$$\boldsymbol{x}_{aug} = \lambda \boldsymbol{x}_{weak} + (1-\lambda)\boldsymbol{x}_{strong}, \text{ where } \lambda \sim \text{Beta}(\alpha, \alpha). \tag{11}$$

## A.16 ANALYSIS OF THRESHOLD

We test the sensitivity of Jump-teaching to the threshold $\tau$. We conduct experiments on *CIFAR-10* with symmetric noise ratio $\epsilon = 0.5$ and asymmetric noise ratio $\epsilon = 0.4$. As shown in Table 11, the results indicate a relatively stable performance with different thresholds. In other words, the effectiveness of Jump-teaching is not significantly influenced by the value of $\tau$ and is more adaptable to various datasets. We attribute the success to two main factors. First, the Hadamard matrix for hashing codes exhibits uniformness in each column (Yang et al., 2015). Second, we utilize classification and auxiliary heads to collaboratively select samples. This guarantees that an adequate quantity of samples is always available for learning. Both of the two factors significantly enhance the stability and accuracy of the model across different datasets.

Table 11: Test Accuracies(%) with different threshold settings.

| $\tau$ | 0.001 | 0.005 | 0.01 | 0.05 | 0.1 |
|---|---|---|---|---|---|
| Sym. $\epsilon = 0.5$ | 92.12 | 92.17 | 92.04 | 91.97 | 92.17 |
| Asym. $\epsilon = 0.4$ | 90.71 | 90.66 | 90.79 | 90.82 | 90.70 |

## A.17 THE CHOICE OF JUMP STEP

In the Table 12, we test the effect of jump steps in Jump-teaching. It is shown that performance with various step settings remains stable. In the Jump-update Strategy, we opt for a step size of 2, obviating the need for additional forward propagation. Furthermore, empirical results indicates a marginal decline in performance as step size increases. This evidence corroborates the assertions in Sec. 3.1. That is to say, delays in updating tables can detrimentally impact the network assimilating newly introduced clean samples.

## A.18 THE STRUCTURES OF BACKBONES AND THE AUXILIARY HEAD

The auxiliary head of Jump-teaching consists of three fully connected layers with ReLU activations and dropout layers between two fully connected layers, followed by a Tanh activation. For *CIFAR-10* and *Clothing1M*, the output dimension of the last fully connected layer is set to 32, whereas setting to 64 for *CIFAR-100*. The intermediate dimension is uniformly set to 512.

---

[*]https://github.com/JackYFL/DISC

[†]https://github.com/Kthyeon/FINE_official

[‡]https://github.com/pxiangwu/TopoFilter

[§]https://github.com/xsshi2015/Self-paced-Resistance-Learning

Table 12: Test accuracy(%) on *CIFAR-10* and *CIFAR-100* with symmetric and asymmetric noise among diverse noise settings.

| Noise type and ratio/ Step | 1 | 2 | 3 | 4 | 5 | 10 | 15 | 20 | 25 | 30 |
|---|---|---|---|---|---|---|---|---|---|---|
| CIFAR-10 Sym. $\epsilon = 0.8$ | 82.81 | 83.37 | 82.47 | 83.09 | 82.55 | 82.04 | 81.03 | 81.35 | 81.46 | 79.77 |
| CIFAR-10 Asym. $\epsilon = 0.4$ | 90.91 | 90.77 | 90.66 | 90.78 | 90.86 | 90.42 | 90.35 | 90.60 | 90.69 | 90.23 |
| CIFAR-100 Sym. $\epsilon = 0.5$ | 66.91 | 67.03 | 66.85 | 66.21 | 66.27 | 65.54 | 65.87 | 65.88 | 65.46 | 64.67 |

### A.19 EXPERIMENTS OF THE JUMP-UPDATE STRATEGY ON THE *Clothing1M* DATASET

In this section, we evaluate the performance of the jump-update strategy on the *Clothing1M* dataset. Specifically, we have integrated the jump-update strategy with Co-teaching and DivideMix. The experimental setup for Co-teaching adheres to Xia et al. (2023), and the setup for DivideMix follows Li et al. (2020). As shown in Table 13, the integration of the jump-update strategy results in performance improvements of 2.88% for Co-teaching and 0.26% for DivideMix, demonstrating the reliability of the jump-update strategy in real-world scenarios.

Table 13: Test accuracy(%) on *Clothing1M*.

| Method | Co-teaching (2*ResNet18) | J-Co-teaching (1*ResNet18) | DivideMix | J-DivideMix |
|---|---|---|---|---|
| Accuracy | 67.94 | **70.82** | 74.17 | **74.43** |