# OpenReview forum: "Jump-teaching: Ultra Robust and Efficient Learning with Noisy Labels"
_ICLR.cc/2025/Conference — Submitted to ICLR 2025_

### Official Review · Reviewer_FvYm · 2024-11-04

**Soundness:** 3
**Presentation:** 3
**Contribution:** 2
**Rating:** 5
**Confidence:** 3

**Summary:**

This paper proposes the Jump-Teaching methodology for learning with noisy labels.

Specifically, it investigates an efficient approach that requires only a single network.

To achieve this, the authors introduce two key techniques: a Jump-update Strategy to mitigate selection bias and Semantic Loss Decomposition to simplify the selection operation.

The effectiveness of the proposed approach is demonstrated through experiments on three benchmark datasets: CIFAR-10, CIFAR-100, and Clothing-1M.

**Strengths:**

Research on sample selection methods using a single network to reduce computational costs is an interesting research topic.

**Weaknesses:**

The academic novelty of this approach appears limited. It is unclear whether updating the model based on selections from the previous step offers any theoretical advantage over the naive approach of updating the model at every iteration. Additionally, is there theoretical support that using data from the previous step effectively addresses the noisy label problem? Moreover, extracting useful information from models at different training epochs has already been extensively explored in the literature. A seminal work in this area, for example, is Snapshot Ensembles: Train 1, get M for free (ICLR ’17).


The experimental results are not convincingly state-of-the-art. In particular, several recent relevant papers (a, b, c) are missing from the references. Their result tables show significantly better performance on CIFAR-10 and CIFAR-100 compared to the results presented in this work.

(a) Generalized Jensen-Shannon Divergence Loss for Learning with Noisy Labels (NeurIPS’21)

(b) DISC: Learning From Noisy Labels via Dynamic Instance-Specific Selection and Correction (CVPR’23)

(c) Sample-wise Label Confidence Incorporation for Learning with Noisy Labels (ICCV’23)

**Questions:**

It is not clear why using data selection results from previous iterations for model updates would be beneficial. Specifically, why would the set of samples selected by the model in the previous step yield better results than the samples selected in the current step? Is the main advantage simply that it avoids sequential updates, thereby reducing the amplification of error accumulation?

---

> ### Author Response · Authors · 2024-11-22
> **Response to Reviewer FvYm (part 1)**
>
> Thank you for your helpful review. We have clarified the novelty of our method in [Novelty]. Furthermore, we add new experiments in [Experiments] and improve the presentation in [Presentation]. Now we would like to address your concerns in detail.
>
> > [W1] The academic novelty of this approach appears limited. [W1.1] It is unclear whether updating the model based on selections from the previous step offers any theoretical advantage over the naive approach of updating the model at every iteration. [W1.2] Additionally, is there theoretical support that using data from the previous step effectively addresses the noisy label problem? [W1.3] Moreover, extracting useful information from models at different training epochs has already been extensively explored in the literature. A seminal work in this area, for example, is Snapshot Ensembles: Train 1, get M for free (ICLR ’17).
>
> Thank you for your comments. We address your concerns point by point to clarify the academic novelty of the jump-update strategy.
>
> **W1.1:** First, there seems to be a misunderstanding of the jump-update strategy. The jump-update strategy also updates the model at every iteration. As for the theoretical advantages over the naive self-update approach, we have provided two main justifications in the paper:
>
> (1) It bridges the disagreement for self-correction of bias, which is shown directly in Figure 2.
>
> (2) It splits the sequential error flow, which can reduce the number of error accumulations by hundreds of times on CIFAR-10 and CIFAR-100, which is detailed in Section 3.1.
>
> **W1.2:** The question appears to be somewhat imprecise. Do you mean that data from the previous step is better than current data? As for this question, the paper does not make such a comparison, nor does it claim any superiority of data from the previous step. If your concern comes from other parts, would you please provide more information? We will provide more details in the following discussion phase.
>
> **W1.3:** The description is overly general, and the referred literature is unrelated to the field of our paper. Therefore, it does not constitute a limitation for the academic novelty of the paper.
> First of all, "extracting useful information from models at different training epochs" is too general, which occurs in countless papers [1, 2, 3, 4, 5, 6, 7, 8, 9, 10]. These papers all follow this manner, but the research problems, research objects, novelty, and technical contributions are diverse but all admirable. We take the paper [11] you referenced as an example.
> - In the research problem, we study how to improve network performance when learning with noisy labels while [11] studies how to improve the ensembling performance of different epochs.
> - In research objects, we study the model update strategy and sample selection criteria while [11] focuses on the convergence and optimization of model parameters.
> -  In temporal-related technologies, we use different selection behaviors to mitigate bias while [11] uses different snapshot parameters for ensembling.
>
> [1] Temporal Ensembling for Semi-Supervised Learning \
> [2] Mean teachers are better role models:  Weight-averaged consistency targets improve  semi-supervised deep learning results  \
> [3] Temporal Self-Ensembling Teacher for Semi-Supervised Object Detection  \
> [4] Averaging Weights Leads to Wider Optima and Better Generalization \
> [5] Linear Combination of Saved Checkpoints Makes Consistency and Diffusion Models Better  \
> [6] Boost Neural Networks by Checkpoints \
> [7] Understanding and Improving Early Stopping for Learning with Noisy Labels \
> [8] Efficient Knowledge Distillation from Model Checkpoints \
> [9] DISC: Learning From Noisy Labels via Dynamic Instance-Specific Selection and Correction \
> [10] Explaining Deep Learning Representations by Tracing the Training Process \
> [11] Snapshot Ensembles: Train 1, get M for free

---

> ### Author Response · Authors · 2024-11-22
> **Response to Reviewer FvYm (part 2)**
>
> > [W2] The experimental results are not convincingly state-of-the-art. In particular, several recent relevant papers (a, b, c) are missing from the references. Their result tables show significantly better performance on CIFAR-10 and CIFAR-100 compared to the results presented in this work.
> (a) Generalized Jensen-Shannon Divergence Loss for Learning with Noisy Labels (NeurIPS’21)
> (b) DISC: Learning From Noisy Labels via Dynamic Instance-Specific Selection and Correction (CVPR’23)
> (c) Sample-wise Label Confidence Incorporation for Learning with Noisy Labels (ICCV’23)
>
> Thank you for the comment, but we cannot fully agree with the comment because the **experimental setup** and **the categories** of referred methods differ from those in our paper.
>
> 1) It is inappropriate to directly compare the values across different tables, as they are based on different experimental settings.
>   - Both (a) and (c) use PreActResNet-34 with 400 epochs, while Jump-teaching uses PreActResNet-18 with only 200 epochs.
>   - Both (a) and (c)  utilize a hyperparameter optimization budget and mechanism, whereas we use fixed hyperparameters.
>
> Consequently, even under identical noise conditions (e.g., symmetric noise of 0.8), their baseline CE implementation achieves a higher accuracy of $39.2\%$, which reflects the advantages of their setup rather than inherent methodological superiority.
>
> 2) It is also inappropriate to compare methods that belong to different categories. Specifically, (a) and (c) are noise-robust loss methods, whereas (b) and (c) employ label correction techniques. Therefore, (a), (b), and (c) all benefit from additional supervision provided by corrupted noise samples. In contrast, Jump-teaching relies solely on clean samples.
>
> If possible, we really recommend reviewing the experiment again and taking care of the experiment setting. After that, we hope the reviewer recognizes the value of our work as demonstrated by the experiments. We have not blindly integrated costly tricks to improve performance, such as double-cost view augmentation, semi-supervised technology, and mature noise-robust functions.  Instead, we focus on optimizing sample selection while also prioritizing training and storage efficiency, which we believe could offer significant value to the community. As shown in Tables 1, 2, and 3, Jump-teaching has achieved state-of-the-art (SOTA) performance in both efficiency and sample selection accuracy, demonstrating its effectiveness in distinguishing noisy samples from clean ones. This provides a solid foundation for future integration with semi-supervised methods such as [1, 2, 3].
>
> [1] FixMatch: Simplifying Semi-Supervised Learning with Consistency and Confidence \
> [2] MixMatch: A Holistic Approach to Semi-Supervised Learning \
> [3] FlexMatch: Boosting Semi-Supervised Learning with Curriculum Pseudo Labeling
>
> > [Q1.1] It is not clear why using data selection results from previous iterations for model updates would be beneficial. Specifically, why would the set of samples selected by the model in the previous step yield better results than the samples selected in the current step? [Q1.2] Is the main advantage simply that it avoids sequential updates, thereby reducing the amplification of error accumulation?
>
> This question partially overlaps with [W1].
>
> **Q1.1:** The set of samples selected in the previous step can't yield better results than the samples selected in the current step. The better results are attributed to the mitigation of selection bias.
>
> **Q1.2:** The selection bias leads to accumulated error in an error flow, the jump-update strategy has an advantage in splitting the sequential error flow into error sub-flows, leading to a smaller degree of accumulated error. Therefore, it mitigates the bias and achieves better results. Notably, the jump-update strategy still enables the network to update sequentially. Regarding accumulated error, our contributions can be summarized as follows:
> - We identify that error accumulation adversely affects performance and formalize the accumulation procedure mathematically.
> - Based on the formulation, we provide some verified methods to improve performance, such as reducing updating frequency in line 268 and avoiding initial bias in Appendix 6.
> - Jump-update strategy offers a novel way to reduce error accumulations by splitting error flows. It not only guarantees sufficient updating frequency but also reduces the number of accumulations to a small magnitude.
>
> [1] Co-teaching: Robust Training of Deep Neural Networks with Extremely Noisy Labels

---

### Official Review · Reviewer_5v7y · 2024-11-04

**Soundness:** 3
**Presentation:** 3
**Contribution:** 3
**Rating:** 5
**Confidence:** 5

**Summary:**

To mitigate compounding selection bias and redundant selection operations in existing methods, the authors of this paper propose a novel framework for optimizing the typical workflow of sample selection, called Jump-teaching. Jump-teaching focuses on discovering significant disagreements within a single network between different training iterations by employing a jump-manner strategy for model updating to bridge the disagreements. Besides, Jump-teaching designs a lightweight plugin to simplify selection operations to help select clean samples more effectively. Finally, experimental results on synthetic and real-world noisy datasets, demonstrate the robustness of Jump-teaching.

**Strengths:**

1. This idea and motivation for discovering significant similarities within a single network between different training iterations are interesting and fascinating.

2. This paper has carried out a lot of formula derivation and proved the effectiveness of the proposed method from the theoretical knowledge level.

3. Figures 1, 2, and 3 in this paper simply and clearly express the main ideas and innovations of the paper.

**Weaknesses:**

1. Authors need to further provide the results of J-Co-teaching and J-DivideMix on Clothing1M in Table 3 to prove the reliable performance in real scenarios.

2. There are errors with the experimental results data. J-Co-teaching does not achieve optimal performance under some settings in Table 4, such as Sym-0.5 of CIFAR-10 and Sym-0.5 and Asym-0.4 of CIFAR100.

3. The content of the semantic loss decomposition part of this paper is not strongly related to the main motivation of this paper, Jump-update Strategy, and is more like an auxiliary trick.

4. There is no clear definition of I_{detection} in Eq. (8).

5. The names of the citation methods are not uniform, such as ' JoCoR ' in the relevant work section and ' JoCor ' in the experimental section.

6. It is recommended to compare the proposed method with 2024 SOTAs.

**Questions:**

See above weaknesses.

---

> ### Author Response · Authors · 2024-11-22
> **Response to Reviewer 5v7y**
>
> Thank you for your constructive review. We have clarified the novelty of our method in [Novelty]. Furthermore, we add new experiments in [Experiments] and improve the presentation by checking the typos and clarifying the definition in [Presentation]. Now we would like to address your concerns in detail.
>
> > [W1] Authors need to further provide the results of J-Co-teaching and J-DivideMix on Clothing1M in Table 3 to prove the reliable performance in real scenarios.
>
> Thank you for your constructive suggestion. We have conducted further experiments on J-Co-teaching and J-DivideMix using the Clothing1M dataset.  The supplemental experiments have been incorporated into Section 4.2 in line 482, with the results detailed in Appendix 19. As shown in Table 13, we conclude that the jump-update strategy demonstrates reliable performance in real-world scenarios, achieving a 2.88% improvement for Co-teaching and a 0.26% improvement for DivideMix.
>
> > [W2] There are errors with the experimental results data. J-Co-teaching does not achieve optimal performance under some settings in Table 4, such as Sym-0.5 of CIFAR-10 and Sym-0.5 and Asym-0.4 of CIFAR100.
>
> Thanks for your helpful suggestion. We have rechecked the experimental results and highlighted the top-performing algorithm for each metric with bold annotation in Table 4.
>
> > [W3] The content of the semantic loss decomposition part of this paper is not strongly related to the main motivation of this paper, Jump-update Strategy, and is more like an auxiliary trick.
>
> Thanks for your comment. Jump-update strategy and semantic loss decomposition are equally crucial to our motivation. In other words, semantic loss decomposition is not merely a trick but a key component of Jump-teaching. We would like to re-emphasize our motivation and clarify the role of semantic loss decomposition.
> 1) Motivation.  The motivation of this paper is to develop a robust and efficient solution for handling noisy labels. Generally, the workflow of traditional sample selection methods involves two key aspects: mitigating selection bias and performing selection operations. Both of these aspects present challenges to achieving robustness and efficiency. First, selection bias inevitably occurs due to the classifier's exposure to noisy data in lines 69 to 81. Second, current selection operations are often redundant lines 82 to 91. Jump-teaching addresses both robustness and efficiency concerns by considering these two aspects in its design. Therefore, the Jump-update strategy and semantic loss decomposition are both essential components of our approach.
> 2) Role. Semantic Loss Decomposition simplifies redundant selection operations by avoiding dataset-wide modeling or batch-wise ranking. Specifically, it utilizes a lightweight plugin to decompose a single loss into a detailed distribution within the loss. By leveraging the memorization effect in this distribution, it becomes easier to detect noise. Therefore, Semantic Loss Decomposition plays a critical role in fulfilling our motivation.
>
> >[W4] There is no clear definition of I_{detection} in Eq. (8).
>
> Thanks for your helpful suggestion. For the sake of completeness, we provide the definition of $I_{detection}$ in line 337.
>
> > [W5] The names of the citation methods are not uniform, such as ' JoCoR ' in the relevant work section and ' JoCor ' in the experimental section.
>
> Thanks for your helpful advice. We have thoroughly reviewed the text for typographical errors and updated the name of the method in [1] as 'JoCoR'.
>
> [1] Combating noisy labels by agreement: A joint training method with co-regularization
>
> > [W6] It is recommended to compare the proposed method with 2024 SOTAs.
>
> Thank you for your thoughtful suggestion. We provide a further comparison between the Jump-teaching and the latest work [1], strengthening the validity of our experimental results. The results are supplemented in Table 1 in line 444. This literature also has been included in the references.  As illustrated in Table 1, Jump-teaching consistently outperforms RML across all noise settings, achieving a remarkable $48.8$% improvement in Sym. $\epsilon = 0.8$ on CIFAR-10 and a $24.3$% improvement in Sym. $\epsilon = 0.8$ on CIFAR-100.
>
> [1] Regroup Median Loss for Combating Label Noise (AAAI'24)

---

> > ### Comment · Reviewer_5v7y · 2024-11-25
> >
> > After reading the author's response and other reviewers' comments, I will reduce my initial score.

---

> > > ### Author Response · Authors · 2024-11-25
> > > **Response to Reviewer 5v7y**
> > >
> > > Thank you for your feedback. We believe our response adequately addressed the key points the reviewers raised. However, it is confusing that the score was reduced without specific reasons outlining the shortcomings of our paper. We remain open to further discussion and would greatly appreciate it if you could elaborate on any specific concerns to help us better address them.

---

### Official Review · Reviewer_ZGND · 2024-11-04

**Soundness:** 3
**Presentation:** 3
**Contribution:** 2
**Rating:** 5
**Confidence:** 4

**Summary:**

The paper proposes "Jump-teaching," a novel framework for robust and efficient learning with noisy labels. By introducing a jump-update strategy and a Semantic Loss Decomposition plugin, the method reduces sample selection bias and enhances efficiency. Experiments show Jump-teaching improves performance over state-of-the-art methods, particularly under high noise conditions, with notable gains in memory efficiency and processing speed.

**Strengths:**

- Proposes an innovative jump-update strategy that significantly reduces selection bias in single-network training.
- Semantic Loss Decomposition provides a lightweight yet effective way to distinguish clean and noisy samples.
- Empirically validated with improved accuracy, efficiency, and robustness across various noise levels and datasets.

**Weaknesses:**

- I disagree with the claim that this work is the first to identify disagreements across different iterations within a single network. Prior studies [1][2] have leveraged these disagreements to distinguish clean samples from corrupted data in training sets. I recommend that the authors revise this claim and include comparisons with these two works.

- In Figure 2, the authors introduce the IoU metric to measure disagreements. Although an explanation of IoU is provided in the appendix, could the authors illustrate what range of IoU values is considered preferable? Because I notice that the IoU value of Jump-update is between the values of self-update and cross-update, the performance of Jump-update is the best (see Figure 2(c,d)).

- Property 1.   (1) I have a question regarding the assumption in Property 1, namely that $N_A$ equals $N_{iterations}$. From past experience, the model often generates biased selections initially, then gradually corrects this bias as performance improves, given moderate noise rates (10%, 20%). Therefore, error accumulation may not persist in later iterations.  (2) Additionally, the results in Figure 4(a) do not align well with the conclusion of Property 1. The highest test accuracy occurs at r = 50% rather than r = 10%.

- There are some concerns regarding whether the jump-update is a more effective strategy for selecting clean samples. (1) In Table 4, at typical noise ratios (e.g., CIFAR-10/100 sym. 50%, CIFAR-100 asym. 40%), J-Co-teaching does not outperform standard Co-teaching (2 networks). (2) While non-trivial improvements are observed in Table 1, these gains do not carry over to a semi-supervised learning setting (see Table 5). In some settings, J-DivideMix is worse than DivideMix.

- The compared methods in Table 1 are outdated. Comparing with more recent works is necessary; for example, ProMix (IJCAI'23).


[1] Late Stopping: Avoiding Confidently Learning from Mislabeled Examples. ICCV'23
[2] Self-Filtering: A Noise-Aware Sample Selection for Label Noise with Confidence Penalization. ECCV'22
[3] ProMix: Combating Label Noise via Maximizing Clean Sample Utility. IJCAI'23

**Questions:**

see Weaknesses.

---

> ### Author Response · Authors · 2024-11-22
> **Response to Reviewer ZGND (Part 1)**
>
> Thanks for your detailed and thoughtful reviews. We have clarified the novelty of our method in [Novelty]. Furthermore, we add new experiments in [Experiments] and improve the presentation in [Presentation]. Now we would like to address your concerns in detail.
>
> > [W1] I disagree with the claim that this work is the first to identify disagreements across different iterations within a single network. Prior studies [1][2] have leveraged these disagreements to distinguish clean samples from corrupted data in training sets. I recommend that the authors revise this claim and include comparisons with these two works.
>
> Thank you for your constructive feedback. We agree that prior studies have leveraged predictions of different iterations to distinguish noisy samples. However, our claim is accurate because the concept of "disagreement" in our paper essentially differs from counterparts in [1 2] in terms of definition, described objects, application, calculation, and focus. We would like to provide a detailed comparison to clarify your concerns.
> - **Definition.** The definition of "disagreement" is different from these terms, such as "Fluctuation" [1] and "First-time k-epoch Learning (FkL)" [2]. As stated in lines 75 to 76, the "disagreement" we refer to is derived from [3]. It denotes the differences in selection behaviors within networks. In contrast, [1] defines a similar concept as "fluctuation," which refers to a sample being classified correctly at one moment but misclassified in the subsequent learning step.  "FkL" in [2] refers to the minimum index of the training epoch that the instance has been predicted to its given label for $k$ consecutive epochs.
> - **Described Object.** The object described by "disagreement" is entirely distinct from others. "Fluctuation" or "FkL" refers to the characteristics of samples, while "disagreement" relates to the characteristics of networks. For example, in [3], the object is two networks, whereas in this paper, the object is the network across different iterations.  "Fluctuation" or "FkL" describes variations in judgments within a single sample.
> - **Application.** "Disagreement" is applied in a completely different way from others. "Disagreement" as a network characteristic, is used to design model update strategies, whereas "fluctuation" and "FkL" as sample characteristics define selection criteria.
> - **Calculation and Focus.** The calculation of each term differs significantly, which stems from their distinct focus. From the focus perspective,  "fluctuation" and "FkL" concern whether a sample is clean, while "disagreement" addresses the potential for correcting selection bias in model updates. Therefore, "Fluctuation" and "FkL" emphasize and calculate consistency between model predictions and labels, whereas "disagreement" is independent of labels and is calculated as the Intersection Over Union (IoU) between selected data sets.
>
> Moreover, the meaning we emphasized through "first" in line 106 has not been discovered and utilized.
>   - We are the first to discover that disagreement within a single network not only exists but also persists during the training, shown in Figure 2(a)(b).
>   - We are the first to discover that disagreement within a single network is significant, which is even larger than two networks, as shown in Figure 2(a)(b).
>   - We are also the first to quantify and visualize "disagreement" by the IoU metric.
>
> We sincerely thank you for your dedicated effort and important suggestions! We have made the following revisions for clarity:
>   - Revise the claim by emphasizing its beneficial attributes for the mitigation of bias in line 107.
>   - Add the definition of "disagreement" clearly in lines 75 and 76. "Disagreement" refers to differences of networks in the selection behaviors.
>   - Add discussions on [1 2] in Related Work in lines 151 and 152.
>   - Add discussions to compare our paper and the related papers in Appendix 2.
>
> [1] Self-Filtering: A Noise-Aware Sample Selection for Label Noise with Confidence Penalization \
> [2] Late stopping: Avoiding confidently learning from mislabeled examples \
> [3] Co-teaching: Robust Training of Deep Neural Networks with Extremely Noisy Labels

---

> ### Author Response · Authors · 2024-11-22
> **Response to Reviewer ZGND (Part 2)**
>
> > [W2] In Figure 2, the authors introduce the IoU metric to measure disagreements. Although an explanation of IoU is provided in the appendix, could the authors illustrate what range of IoU values is considered preferable? Because I notice that the IoU value of Jump-update is between the values of self-update and cross-update, the performance of Jump-update is the best (see Figure 2(c,d)).
>
> Thank you for your response. It is both interesting and beneficial to discuss this observation. The pattern should be the smaller the IOU values (the larger the disagreement), the better the performance.
> 1. However, as you noticed, the performance of Self-update is poorer than Jump-update and Cross-update despite the smallest IoU values, which does not suit this pattern. It is because there is an underlying assumption that disagreement should have been exploited but Self-update does not utilize this advantage.
> 2. Moreover, Jump-update and Self-update both show very small IOU values, with Jump-update slightly larger than Self-update. This is because the Jump-update strategy bridges disagreement, so the IoU values increase.
> Appendix 1 has been supplemented by focusing on the analysis of the relationship between IoU and performance in three strategies.
>
> > [W3] Property 1. (1) I have a question regarding the assumption in Property 1, namely that NA equals Niterations. From past experience, the model often generates biased selections initially, then gradually corrects this bias as performance improves, given moderate noise rates (10%, 20%). Therefore, error accumulation may not persist in later iterations. (2) Additionally, the results in Figure 4(a) do not align well with the conclusion of Property 1. The highest test accuracy occurs at r = 50% rather than r = 10%.
>
> Thanks for your positive and thoughtful suggestion.
>
> For (1), the reviewer questioned that the bias may also be corrected by performance improvement. There is a possibility of that occurring. Moreover, Stochastic Gradient Descent also could lead to $N_A$ not being equal to $N_{iterations}$. Therefore, we revised the paper by adding a hypothesis that the error flow is an uninterrupted model in lines 227 and 768. The detailed proof of $N_A$ equals $N_{iterations}$ can be found in Appendix 4. Notably, $N_A$ equals $N_{iterations}$ under the ideal condition, when we consider the two factors above, Property 1 still holds because $N_A$ is still proportional to $N_{iterations}$, considering corrections caused by either performance improvement or SGD are random.
>
> For (2), this conclusion is right. This issue arises from the fact that accuracy as the experimental result cannot be fully aligned with the accumulated error $D_A$. Also, it is affected by other factors in the experiment. As illustrated in lines 796 and 797, if the model cannot be trained quickly, performance improvements will be delayed. What makes things worse is the selection does not become accurate immediately, leading to a long-term fit to noise. This may explain why the highest test accuracy occurs at $r = 50\%$ rather than $r = 10\%$.
>
> Regarding Property 1, this conclusion remains intuitive. Considering other factors, the accuracy can roughly reflect $D_A$ with fluctuations. Moreover, both $r = 10\%$ and $r = 50\%$ show significant improvements in accuracy, effectively validating Property 1.

---

> ### Author Response · Authors · 2024-11-22
> **Response to Reviewer ZGND (Part 3)**
>
> > [W4] There are some concerns regarding whether the jump-update is a more effective strategy for selecting clean samples. (1) In Table 4, at typical noise ratios (e.g., CIFAR-10/100 sym. 50%, CIFAR-100 asym. 40%), J-Co-teaching does not outperform standard Co-teaching (2 networks). (2) While non-trivial improvements are observed in Table 1, these gains do not carry over to a semi-supervised learning setting (see Table 5). In some settings, J-DivideMix is worse than DivideMix.
>
> Thank you for your concerns. Overall, the jump-update strategy is more effective and efficient, especially in extreme noise settings.
> - For (1)
>   - Under typical noise levels, the performance of the jump-update strategy is comparable to that of co-teaching. Both methods exhibit higher accuracy in certain noise scenarios, with the accuracy difference not exceeding one point. (In this case, different seeds might lead to varying results.) Under extreme noise (noise rate =0.8, 0.9), its performance significantly outperforms co-teaching.
>   - The jump-update only employs a single network with higher efficiency.
> - For (2)
>   - In most scenarios, J-DivideMix outperforms DivideMix. Furthermore, under all extreme noise conditions (90% symmetric noise), its performance is significantly superior to that of DivideMix. As a model update strategy, J-DivideMix aims to minimize selection bias, achieving optimal results even in extreme noise situations. Regarding the integration with semi-supervised methods, this remains a long-term challenge. It involves not only the effective use of unlabeled samples but also balancing efficiency and robustness. In this paper, we primarily focus on supervised-only scenarios. We are committed to making dedicated efforts in the future to address these limitations.
>
>
> > [W5] The compared methods in Table 1 are outdated. Comparing with more recent works is necessary; for example, ProMix (IJCAI'23).
>
> Thank you for your insightful advice. We have replicated and supplemented the results of the latest sample selection method [1], which are presented in Table 1 of the revised paper. Although the least method employs loss estimation to further protect the model from noisy samples, it can not mitigate the compounding selection bias. Therefore, Jump-teaching still achieves state-of-the-art performance in all the noise settings. As shown in Table 1, Jump-teaching achieves a remarkable 48.8% higher accuracy in Sym. $\epsilon = 0.8$ on CIFAR-10 and a 24.3% higher accuracy in Sym. $\epsilon = 0.8$ on CIFAR-100. Table 1 shows the comparison results in supervised-only scenarios, therefore we do not include ProMix.
>
> [1] Regroup Median Loss for Combating Label Noise (AAAI'24)

---

### Official Review · Reviewer_ySnh · 2024-11-04

**Soundness:** 1
**Presentation:** 1
**Contribution:** 2
**Rating:** 5
**Confidence:** 3

**Summary:**

The authors introduce the concept of identifying significant disagreements within a single neural network across different training iterations. This discovery leads to the proposal of a "jump-manner" strategy for model updates, effectively bridging the gaps caused by these disagreements. Jump-Teaching simplifies the sample selection process through a lightweight plugin that generates a clear loss distribution in an auxiliary encoding space. This approach enhances the ability to select clean samples more effectively, addressing selection bias and redundancy. The framework demonstrates substantial improvements in both robustness and efficiency compared to state-of-the-art methods, specifically reducing peak memory usage by 46% and increasing training speed by up to 253%. The paper highlights that current methods can benefit from integrating with the Jump-Teaching framework, suggesting that it enhances the overall approach to learning with noisy labels.

**Strengths:**

1. The method achieves improved performance under high noise rates on CIFAR datasets.
2. The method demonstrates better computational and storage efficiencies during testing.

**Weaknesses:**

1. The experiments conducted on CIFAR-10 with 90% symmetric noise lack meaningful insight, as this setting results in random labels for each sample, effectively reducing the task to an unsupervised learning scenario.
2. The presentation needs improvement. Suggested changes include:
   - The methodology section incorporates experimental analysis (Figure 4), making it difficult to discern insights related to debiasing.
   - The connection among the four subsections in Section 3.2 is unclear.
   - The framework presented in Figure 1 contains excessive details that are not explained in the introduction; these should either be removed or relocated to the methodology section.
3. The authors claim that “Jump-Teaching is the first work to discover significant disagreements within a single network between different training iterations.” However, the concept of leveraging disagreements across different training iterations has been previously studied (see [1]).

[1] Self-Filtering: A Noise-Aware Sample Selection for Label Noise with Confidence Penalization, ECCV 2022.

**Questions:**

1. As indicated in Table 5, the accuracy under 90% symmetric noise on CIFAR-10 exceeds 75%, which corresponds to random labels for the training samples. This scenario can be classified as an unsupervised learning task rather than weakly supervised learning. We need to reconsider the implications of generalization in learning with noisy labels using semi-supervised learning methods, given the lack of supervision.
2. It seems unreasonable to separate the updates of the neural network parameters in steps 9-10. Combining \( L^{BCE} \) and \( L^{CE} \) and updating the neural network with respect to the total loss could be more efficient.

---

> ### Author Response · Authors · 2024-11-22
> **Response to Reviewer ySnh (Part 1)**
>
> Thanks for your thorough reviews. We have clarified the novelty of our method in [Novelty]. Furthermore, we add new experiments in [Experiments] and improve the presentation in [Presentation]. Now we would like to address your concerns in detail.
>
> > [W1] The experiments conducted on CIFAR-10 with 90% symmetric noise lack meaningful insight, as this setting results in random labels for each sample, effectively reducing the task to an unsupervised learning scenario.
>
> There may be a misunderstanding that 90% is mistaken as the ratio of flipped categories. Actually, 90% symmetric noise is advisable for the following three reasons:
> 1) The setup that uses CIFAR-10 with 90% symmetric noise is aligned with the baseline [1] in Table 5.
> 2) 90% symmetric noise serves as a widely adopted standard [2, 3, 4, 5] for performance evaluation under extreme noise.
> 3) The setup is reasonable, as the labels are not random, indicating that this task is not an unsupervised learning scenario. We conduct experiments detailing that the labels of the selected samples are not random at all. Specifically, we calculate the ratio and number of clean samples selected by the network under 90% symmetric noise conditions. As shown in the newly added Fig. 8, the clean ratio by all methods exceeded 20%, and even two methods using the Jump-update strategy exceeded 60%, which is significantly higher than the random proportion.
>
> In fact, 90% noise indicates that a label has a 90% probability of being flipped to any class. We also recommend that readers refer to Appendix 12 for further details on the simulation of synthetic noise.
>
> [1] DivideMix: Learning with Noisy Labels as Semi-supervised Learning \
> [2] LongReMix: Robust Learning with High Confidence Samples in a Noisy Label Environment \
> [3] Sample Prior Guided Robust Model Learning to Suppress Noisy Labels \
> [4] Probabilistic End-to-end Noise Correction for Learning with Noisy Labels \
> [5] Joint Optimization Framework for Learning with Noisy Labels
>
> > [W2.1] The methodology section incorporates experimental analysis (Figure 4), making it difficult to discern insights related to debiasing.
>
> The insights presented in the Empirical Analysis (the second subsection) are verified by Experimental Analysis (the third subsection). We have revised lines 186 and 187 to point out the relationship between the two explicitly.
> In the Methodology section (Section 3.1), we first describe the model update strategy for debiasing in the first subsection, followed by a discussion of our insights in the second subsection. The final subsection, titled 'Experimental Analysis', presents experimental results that validate these insights.
> In Experimental Analysis subsection, we introduce the experimental settings in the first paragraph and use the second and third paragraphs to verify the two core properties underlying the insights.
> The structure of the section, with clear subsection titles and introductory sentences, makes it easy for readers to follow. If there are any specific points of confusion, please feel free to let us know, and we will be happy to clarify.
>
> > [W2.2] The connection among the four subsections in Section 3.2 is unclear.
>
> Thank you for your constructive review! The unclarity stems from the lack of a clear guide for the structure of these subsections. We have explicitly explained the relationship between these sections in lines 299 and 300.
> Section 3.2 is organized as follows: we first introduce our motivation in the first subsection, followed by detailed explanations of the codebook and auxiliary head in the next two subsections. Finally, we describe the sample selection operation that leverages both modules in the fourth subsection.
>
> > [W2.3] The framework presented in Figure 1 contains excessive details that are not explained in the introduction; these should either be removed or relocated to the methodology section.
>
> The concern stems from two aspects. First, Figure 1 contains excessive details, which create a barrier to reading. Second, Figure 1 was not effectively integrated into the introduction.
> To address these issues, we have made two revisions:
> - We simplify Figure 1 by removing unnecessary elements, making it more intuitive and better aligned with the introduction.
> - We revise lines 98 and 99 to explicitly guide readers to refer to Figure 1 for a clearer understanding of the content.

---

> ### Author Response · Authors · 2024-11-22
> **Response to Reviewer ySnh (Part 2)**
>
> > [W3] The authors claim that “Jump-Teaching is the first work to discover significant disagreements within a single network between different training iterations.” However, the concept of leveraging disagreements across different training iterations has been previously studied (see [1]).
>
> Thank you for your thorough review.  Our method differs significantly from others, and there may be a misunderstanding stemming from the similarity in that our paper and [2, 3] both leverage network inferences across different iterations. However, it is not a new thing that predictions change in different iterations. Therefore, we will conduct a detailed and clear exposition to eliminate misunderstandings and avoid ambiguity through a series of revisions.
>
> Although "disagreement" and "fluctuation" are similar in form, they are essentially different:
> - **Definition.** The definition of "disagreement" is different from "fluctuation". As stated in lines 75 and 76, the "disagreement" we refer to is derived from [1]. It denotes the differences in selection behaviors within networks. In contrast, "fluctuation" refers to a sample being classified correctly at one moment but misclassified in the subsequent learning step.
> - **Described Object.** The object described by "disagreement" is entirely distinct from "fluctuation". "Fluctuation" refers to the characteristics of samples, while "disagreement" relates to the characteristics of networks. For example, in [1], the object is two networks, whereas in this paper, the object is the network across different iterations.  "Fluctuation" describes variations in judgments within a single sample.
> - **Application.** "Disagreement" is applied in a completely different way from "fluctuation". "Disagreement" as a network characteristic, is used to design model update strategies, whereas "fluctuation" as sample characteristics define selection criteria.
> - **Calculation and Focus.** The calculation of each term differs significantly, which stems from their distinct focus. From the focus perspective,  "fluctuation" concerns whether a sample is clean, while "disagreement" addresses the potential for correcting selection bias in model updates. Therefore, "Fluctuation" emphasizes and calculates consistency between model predictions and labels, whereas "disagreement" is independent of labels and is calculated as the Intersection Over Union (IoU) between selected data sets.
>
> In addition to the differences between the two in terms of definition, described object, application, computation, and focus, we would also like to emphasize that the original expression in line 106 contains several meanings regarding "first":
> - We are the first to discover that disagreement within a single network not only exists but also persists during the training, shown in Figure 2(a)(b).
> - We are the first to discover that disagreement within a single network is significant, which is even larger than two networks, as shown in Figure 2(a)(b).
> - We are also the first to quantify and visualize "disagreement" by the IoU metric.
> We are motivated by this discovery of existence, persistence, and significance to design an efficient and effective jump-update strategy. Before our paper, the disagreement that the field is generally considered only exists in two networks.
>
> We sincerely thank you for your dedicated effort and respect for the great work that has been done in [2]. We have made the following revisions for clarity:
> - Revise the claim by emphasizing its beneficial attributes for the mitigation of bias in line 107.
> - Add the definition of "disagreement" clearly in lines 75 and 76. "Disagreement" refers to differences of networks in the selection behaviors.
> - Add discussions on [2,3] in Related Work in lines 51 and 152.
> - Add discussions to compare our paper and the related papers in Appendix 2.
>
> [1] Co-teaching: Robust Training of Deep Neural Networks with Extremely Noisy Labels \
> [2] Self-Filtering: A Noise-Aware Sample Selection for Label Noise with Confidence Penalization \
> [3] Late stopping: Avoiding confidently learning from mislabeled examples

---

> ### Author Response · Authors · 2024-11-22
> **Response to Reviewer ySnh (Part 3)**
>
> > [Q1] As indicated in Table 5, the accuracy under 90% symmetric noise on CIFAR-10 exceeds 75%, which corresponds to random labels for the training samples. This scenario can be classified as an unsupervised learning task rather than weakly supervised learning. We need to reconsider the implications of generalization in learning with noisy labels using semi-supervised learning methods, given the lack of supervision.
>
> The question is similar to [W1]. 90% symmetric noise on CIFAR-10 is not a part of unsupervised learning scenarios because the labels of selected samples are not random. The details can be referred to in response to W1.
> Moreover, the accuracy under 90% symmetric noise on CIFAR-10 exceeds 75%, demonstrating the potential of weakly supervised learning methods with insufficient supervision. However, it is also crucial to consider the importance of sample selection to separate labeled samples and unlabeled samples from noisy data.
>
> > [Q2] It seems unreasonable to separate the updates of the neural network parameters in steps 9-10. Combining ( L^{BCE} ) and ( L^{CE} ) and updating the neural network with respect to the total loss could be more efficient.
>
> Thanks for your helpful suggestion. These two steps are updated simultaneously. We revise Algorithm 1 in line 358. We combine the two steps in the code directly and backpropagate them simultaneously, ensuring no impact on efficiency. In the former version of Algorithm 1, we separated them only for clarity.

---

> > ### Comment · Reviewer_ySnh · 2024-11-27
> >
> > Thank you for the detailed response, which addresses most of my concerns. Based on the clarifications provided, I am inclined to increase my score to 5. However, I still believe that the paper falls slightly below the acceptance threshold.

---

### Author Response · Authors · 2024-11-22
**Response to All Reviewers (Part 2)**

**[Experiments]**

R.ZGND and R.5v7y recommended comparing our work with recent studies. In response, we expand the following experiments to support our paper further:
- **Supplementing baselines in Table 1.** To compare with the latest method, we include the results of RML [4] in Table 1 to showcase the up-to-date effectiveness of Jump-teaching.
- **Statistical analysis of labels under 90% symmetric noise.** We conduct experiments to calculate the ratio and number of selected labels under 90% symmetric noise. The results confirm that the selected labels are not random, thereby validating the rationality of our experimental setup.
- **Conducting additional experiments on real-world datasets.** We conduct further experiments using J-Co-teaching and J-DivideMix on the Clothing1M dataset to demonstrate the reliable performance of the Jump-update strategy in real-world scenarios.

[1] Self-Filtering: A Noise-Aware Sample Selection for Label Noise with Confidence Penalization \
[2] Late stopping: Avoiding confidently learning from mislabeled examples \
[3] Co-teaching: Robust Training of Deep Neural Networks with Extremely Noisy Labels \
[4] Regroup Median Loss for Combating Label Noise (AAAI'24)


**[Presentation]**

We have improved our presentation by:
- Simplify Figure 1 to better align with the introduction.
- Enhance the connection between Figure 1 and the text in lines 98 and 99.
- Add necessary explanations, such as disagreement and the notation $I_{detection}$.
- Revise Property 1 by incorporating a hypothesis.
- Correct a few typos.

---

### Author Response · Authors · 2024-11-22
**Response to All Reviewers (Part 1)**

We sincerely thank all the reviewers for their valuable time and insightful suggestions. In this 'global' rebuttal, we clarify our technical contributions and present new experimental results. Furthermore, we will address other concerns in our responses to individual reviews. A revised version of our paper has been uploaded, with all the modifications marked in blue to address the concerns.

**[Novelty]**

R.Ysnh and R.ZGND expressed concerns regarding our claim that "We are the first work to discover significant disagreement within a single network," as prior studies [1, 2] have utilized predictions of different epochs. This concern can be addressed with three key points:

1)  The concept of "disagreement" in our paper essentially differs from counterparts in [1, 2] in terms of definition, described objects, application, calculation, and focus.
  - **Definition.** The definition of "disagreement" is different from these terms, such as "Fluctuation" [1] and "First-time k-epoch Learning (FkL)" [2]. As stated in lines 75 to 76, the "disagreement" we refer to is derived from [3]. It denotes the differences in selection behaviors within networks. In contrast, [1] defines a similar concept as "fluctuation," which refers to a sample being classified correctly at one moment but misclassified in the subsequent learning step.  "FkL" in [2] refers to the minimum index of the training epoch that the instance has been predicted to its given label for $k$ consecutive epochs.
  - **Described Object.** The object described by "disagreement" is entirely distinct from others. "Fluctuation" or "FkL" refers to the characteristics of samples, while "disagreement" relates to the characteristics of networks. For example, in [3], the object is two networks, whereas in this paper, the object is the network across different iterations.  "Fluctuation" or "FkL" describes variations in judgments within a single sample.
  - **Application.** "Disagreement" is applied in a completely different way from others. "Disagreement" as a network characteristic, is used to design model update strategies, whereas "fluctuation" and "FkL" as sample characteristics define selection criteria.
  - **Calculation and Focus.** The calculation of each term differs significantly, which stems from their distinct focus. From the focus perspective,  "fluctuation" and "FkL" concern whether a sample is clean, while "disagreement" addresses the potential for correcting selection bias in model updates. Therefore, "Fluctuation" and "FkL" emphasize and calculate consistency between model predictions and labels, whereas "disagreement" is independent of labels and is calculated as the Intersection Over Union (IoU) between selected data sets.

2) The meaning we emphasized through "first" has not been discovered and utilized.
  - We are the first to discover that disagreement within a single network not only exists but also persists during the training, shown in Figure 2(a)(b).
  - We are the first to discover that disagreement within a single network is significant, which is even larger than two networks, as shown in Figure 2(a)(b).
  - We are also the first to quantify and visualize "disagreement" by the IoU metric.
3) We sincerely thank all the reviewers for their dedicated efforts. We have made the following revisions for further clarity:
  - Revise the claim by emphasizing its beneficial attributes for the mitigation of bias in line 107.
  - Add the definition of "disagreement" clearly in lines 75 and 76. "Disagreement" refers to differences of networks in the selection behaviors.
  - Add discussions on [1, 2] in Related Work in lines 151 and 152.
  - Add discussions to compare our paper and the related papers in Appendix 2.

---

### Author Response · Authors · 2024-12-02
**Summary of Submission and Discussion**

We sincerely thank the reviewers and chairs for their valuable feedback. During the discussion period, we made every effort to address all of the reviewers' concerns, particularly regarding the novelty and effectiveness of the proposed method. Additionally, we have conducted the suggested experiments and made several revisions, all of which are highlighted in blue in the uploaded PDF file. Although we are still awaiting responses from some reviewers, we would be happy to provide a clear summary of the paper and the points discussed.

## Motivation
This paper focuses on combating noisy labels by selecting clean samples for training. The motivation behind Jump-teaching is to optimize the outdated sample selection workflow by mitigating selection bias within a single network and simplifying redundant selection operations through the incorporation of a lightweight plugin.
## Contributions
- **Novelty of Discovery.** We are the first work to discover the disagreement for correcting selection bias within a single network. Notably, this has been clarified in the [Novelty] section of the general response and in specific responses to Reviewer #1 and Reviewer #2.
- **Technical Contribution of Jump-update Strategy.** Jump-update strategy proposes a simple and cost-free solution to bridge disagreement and split sequential error flows, leading to a significantly better trade-off between efficiency and robustness.
- **Technical Contribution of Semantic Loss Decomposition.** The proposed plugin virtually eliminates redundancy in selection operations, occupying only 2.2% of the total training time, by exploiting the memorization effects in the distribution of the decomposed single loss.
## Experiments
Extensive experimental results confirm the effectiveness of our method.
- **Superior Performance of Jump-teaching.** To address the concerns regarding up-to-dateness, we compare 2024 SOTA in Table 1. Jump-teaching outperforms the latest baselines across various noise ratios and types, including symmetric noise (Table 1), asymmetric noise (Table 1), pairflip noise (Table 9), IDN (Table 10), and real-world noise (Table 3). Furthermore, it reduces peak memory usage by $0.46\times$ and accelerates training speed by up to $2.53\times$ (Table 2), achieving an extremely low overhead for sample selection (2.2% in training speed and 2.8% in peak memory).
- **Effective Integration of Jump-update Strategy.** We successfully apply the Jump-update strategy to improve the performance of mitigating bias in two representative settings: supervised-only and semi-supervised methods. The results demonstrate impressive improvements under extreme noise conditions (Table 4 and Table 5) and real-world noise (Table 13).  The newly added results in Table 13 also address Reviewer #3's concerns regarding the performance of J-co-teaching and J-DivideMix under real-world noise.
We also summarize the newly conducted experiments addressing these concerns in the [Experiments] section of the general response and respond to specific reviewers in detail.

---

### Meta-Review · Area_Chair_i3ce · 2024-12-11

**Metareview:**

This paper proposes a novel technique called Jump-teaching for learning with noisy labels. Specifically, Jump-teaching aims to discover significant disagreements within a single network between different training iterations. Based on this discovery, this paper proposes a jump-manner strategy for model updating to bridge the disagreements. The authors further illustrate the effectiveness from the perspective of error flow. Moreover, Jump-teaching designs a lightweight plugin to simplify selection operations. It creates a detailed yet simple loss distribution on an auxiliary encoding space, which helps select clean samples more effectively.

Although the authors claim that their method is novel, the reviewers consider that utilizing the disagreements for sample selection across different iterations within a single network or across different networks is not new, so all of them show negative scores.

**Additional Comments On Reviewer Discussion:**

All reviewers show negative scores to this paper, as the idea of this paper, namely utilizing disagreement, is not interesting enough. Therefore, I feel sorry that I cannot recommend an acceptance to this paper.

---

### Decision · Program_Chairs · 2025-01-22

Reject